# Cepharanthine analogs mining and genomes of *Stephania* accelerate anti-coronavirus drug discovery

Liang Leng [1,10], Zhichao Xu [2,10], Bixia Hong[3,10], Binbin Zhao [4,10], Ya Tian[2], Can Wang[1], Lulu Yang[1], Zhongmei Zou[5], Lingyu Li[5], Ke Liu[3], Wanjun Peng[4], Jiangning Liu[4], Zhoujie An[2], Yalin Wang[2], Baozhong Duan[6], Zhigang Hu[7], Chuan Zheng[8], Sanyin Zhang[1], Xiaodong Li[9], Maochen Li [3], Zhaoyu Liu [1], Zenghao Bi[1], Tianxing He[1], Baimei Liu[1], Huahao Fan [3] ✉, Chi Song [1] ✉, Yigang Tong [3] ✉ & Shilin Chen [1] ✉

Cepharanthine is a secondary metabolite isolated from *Stephania*. It has been reported that it has anti-conronaviruses activities including severe acute respiratory syndrome coronavirus-2 (SARS-CoV-2). Here, we assemble three *Stephania* genomes (*S. japonica*, *S. yunnanensis*, and *S. cepharantha*), propose the cepharanthine biosynthetic pathway, and assess the antiviral potential of compounds involved in the pathway. Among the three genomes, *S. japonica* has a near telomere-to-telomere assembly with one remaining gap, and *S. cepharantha* and *S. yunnanensis* have chromosome-level assemblies. Following by biosynthetic gene mining and metabolomics analysis, we identify seven cepharanthine analogs that have broad-spectrum anti-coronavirus activities, including SARS-CoV-2, Guangxi pangolin-CoV (GX_P2V), swine acute diarrhoea syndrome coronavirus (SADS-CoV), and porcine epidemic diarrhea virus (PEDV). We also show that two other genera, *Nelumbo* and *Thalictrum*, can produce cepharanthine analogs, and thus have the potential for antiviral compound discovery. Results generated from this study could accelerate broad-spectrum anti-coronavirus drug discovery.

The 21st century's first two decades have seen three remarkable coronavirus outbreaks: the severe acute respiratory syndrome coronavirus (SARS-CoV), the Middle East respiratory syndrome coronavirus (MERS-CoV), and the severe acute respiratory syndrome coronavirus 2 (SARS-CoV-2). Anti-coronavirus drugs, especially broad-spectrum anti-coronavirus drugs, are urgently needed; however, drug discovery for this purpose is complicated by constantly emerging SARS-CoV-2 variants and other potential human health-threatening coronaviruses[1,2]. The conventional route to drug discovery, which involves significant time and capital investment, struggles to meet the

[1]Institute of Herbgenomics, Chengdu University of Traditional Chinese Medicine, Chengdu 611137, China. [2]College of Life Science, Northeast Forestry University, Harbin 150040, China. [3]College of Life Science and Technology, Beijing University of Chemical Technology, Beijing 100029, China. [4]NHC Key Laboratory of Human Disease Comparative Medicine, Institute of Laboratory Animal Science, Chinese Academy of Medical Sciences and Comparative Medicine Center, Peking Union Medical College, Beijing 100730, China. [5]Institute of Medicinal Plant Development, Chinese Academy of Medical Sciences & Peking Union Medical College, Beijing 100193, China. [6]College of Pharmaceutical Science, Dali University, Dali 671000, China. [7]College of Pharmacy, Hubei University of Chinese Medicine, Wuhan 430065, China. [8]Hospital of Chengdu University of Traditional Chinese Medicine, Chengdu 610072, China. [9]Wuhan Botanical Garden, Chinese Academy of Sciences, Wuhan 430074, China. [10]These authors contributed equally: Liang Leng, Zhichao Xu, Bixia Hong, Binbin Zhao. ✉e-mail: fanhuahao.1987@163.com; songchi@cdutcm.edu.cn; tongyigang@mail.buct.edu.cn; slchen@cdutcm.edu.cn

demand for broad-spectrum antiviral drug development. Conversely, drug repurposing holds promise in significantly reducing development time, as prior preclinical testing and safety assessments have already been completed[3,4]. Cepharanthine (**1**), a secondary metabolite derived from *Stephania*, is primarily used to boost white blood cell counts following radiotherapy or chemotherapy in patients with cancer. It has garnered attention for its reported inhibition of infection caused by several important human viruses, including SARS-CoV-2[5–8], as well as other viruses such as SARS-CoV, MERS-CoV[9], human immunodeficiency virus type 1 (HIV-1)[10], Ebola virus (EBOV), and Zika virus (ZIKV)[8].

Cepharanthine (**1**) belongs to benzylisoquinoline alkaloids (BIAs), a group of nearly 2,500 specialized plant metabolites with remarkable pharmacological effects[11]. Cepharanthine (**1**), approved by the Pharmaceuticals and Medical Devices Agency (PMDA) in Japan, is used to treat cancer and inflammation. The well-known antibacterial and hypolipidemic compound berberine (**2**)[12] is approved by PMDA and the China Food and Drug Administration (CFDA) to treat infection and parasitology. Tetrandrine (**3**)[13] has been approved by CFDA as a calcium channel blocker for inflammatory disease treatment. The biosynthesis of cepharanthine (**1**) in *Stephania* begins with the condensation of dopamine (**4**) and 4-hydroxyphenylacetaldehyde (4-HPAA, **5**) through norcoclaurine synthase (NCS)[14,15], yielding norcoclaurine (**6**). **6** produces (*S*)-*N*-methycoclaurine (**7**), a reaction catalyzed by 6-*O*-methyltransferase (6OMT) and coclaurine *N*-methyltransferase (NMT), which is a gateway to the generation of various compounds such as protoberberine, morphinan, aporphine, and BIA dimers[16–18]. BIA dimers such as cepharanthine (**1**), formed through oxidative coupling, involve two *N*-methylcoclaurine units in either the (*R*) or (*S*) configuration, are selectively and oxidatively coupled with the oxidase enzyme CYP80A at their benzylic moieties. While the biosynthesis of guattegaumerine (**8**) and berbamunine (**9**) in yeast has been achieved[18,19]. Downstream biosynthetic pathways of BIA dimers including cepharanthine (**1**), tetrandrine (**3**), and berbamine (**10**) remain largely unexplored. Given the potent anti-SARS-CoV-2 property of cepharanthine (**1**), investigating whether its analogs have antiviral activity would be helpful for new broad-spectrum antiviral drug development[20].

In this work, we assemble high-quality *Stephania* genomes, propose a cepharanthine (**1**) biosynthesis pathway, systematically investigate the anti-coronavirus activity of metabolites along this pathway, and deliberate on avenues for leveraging chemodiversity to advance drug discovery.

## Results

### *Stephania* genomes assembly

Through *k*-mer analysis (*k* = 19), we estimated genome sizes for *S. japonica*, *S. yunnanensis*, and *S. cepharantha* to be approximately 636.6, 813.6, and 783.0 Mb, respectively. Heterozygosity ratios ranged from 0.93 to 1.42 (Supplementary Table 1, Supplementary Fig. 1). Based on these findings, we generated 266.9 Gb of nanopore ultralong reads (N50 > 50 kb; genome coverage >100× for each species) and 401.9 Gb DNBSEQ short reads ( > 168× for each species). This enabled assembly and refinement of contig-level assemblies (Table 1, Supplementary Table 2, Fig. 1), yielding three draft genomes with contig N50 values of 22.3, 37.6, and 6.8 Mb (Supplementary Table 3). By applying Hi-throughput chromatin conformation capture (Hi-C)-based scaffolding, we improved the final assembly, resulting in genome sizes of 643.4, 812.2, and 743.5 Mb, and N50 values of 54.0, 56.6, and 50.7 Mb, respectively. Benchmarking universal single-copy ortholog (BUSCO) assessments showed completeness of these assemblies ranging from 96.0% to 96.9% (Table 1, Supplementary Table 4). We predicted 26,742, 32,039, and 31,150 protein-coding genes (Table 1, Supplementary Table 5, Supplementary Figs. 2 and 3). While *S. japonica* had fewer genes than *S. yunnanensis* and *S. cepharantha*, the lengths of the gene (mRNA and CDS), exon, and intron displayed similar distributions

**Table 1 | Summary of sequencing, genome assembly and annotation**

| | | *S. japonica* | *S. yunnanensis* | *S. cepharantha* |
|---|---|---|---|---|
| **Sequencing** | | | | |
| Total base (Gb) | ONT | 97.5 | 81.2 | 88.2 |
| | DNBSEQ | 118.2 | 158.7 | 125.0 |
| | Hi-C | 116.7 | 106.8 | 136.6 |
| N50 (Kb)* | ONT | 51.5 | 51.5 | 53.0 |
| Depth (×) | ONT | 151.5 | 100.0 | 118.7 |
| | DNBSEQ | 183.7 | 195.4 | 168.1 |
| | Hi-C | 181.4 | 131.6 | 183.8 |
| **Assembly** | | | | |
| | Total length (Mb) | 643.4 | 812.2 | 743.5 |
| | Scaffold N50 (Mb) | 54.0 | 56.6 | 50.7 |
| | Longest contig length (Mb) | 86.8 | 69.0 | 67.7 |
| | GC ratio (%) | 37.2 | 36.1 | 36.6 |
| | Number of gaps | 1 | 4 | 11 |
| | BUSCO completeness (%) | 96.9 | 96.8 | 96.0 |
| **Annotation** | | | | |
| | Total number of genes | 26,742 | 32,039 | 31,150 |
| | Average mRNA length (bp) | 6,423.8 | 6,724.2 | 6,297.3 |
| | Average CDS length (bp) | 1,068.3 | 961.8 | 938.2 |
| | Total repeat length (Mb) | 477.4 | 626.6 | 573.3 |
| | Fraction of repeats (%) | 74.2 | 77.2 | 77.2 |
| | BUSCO completeness (%) | 96.8 | 96.1 | 96.2 |

*Read length of DNBSEQ and Hi-C sequencing are both 150 bp.

across the three genomes (Supplementary Table 5, Supplementary Fig. 2). Repeat sequences constituted 74.2%, 77.2%, and 77.2% of the genomes (Table 1, Supplementary Table 6). The BUSCO completeness for annotations ranged from 96.1% to 96.8%, closely mirroring that of genome assemblies (Table 1, Supplementary Table 4).

Orthologous gene families among the three *Stephania* species and 11 other flowering plant species were classified, leading to a phylogenetic tree constructed from 268 single-copy orthologs (Fig. 1A, Supplementary Fig. 4). We determined that *S. cepharantha* and *S. yunnanensis* diverged around 10.1 million years ago (Mya), with *S. japonica* diverging from them approximately 24.8 Mya (Fig. 1A). By employing genome-wide all-against-all Hi-C interaction maps and manual inspection, we observed 11 pairs of chromosomes (2n = 2x = 22) in *S. japonica* genome and 13 pairs (2n = 2x = 26) each in the genomes of *S. cepharantha* and *S. yunnanensis* (Fig. 1B). Utilizing nanopore ultralong reads, we filled gaps, identified the telomere region by telomere repeat (CCCTAAA)-guided search and focal telomere region assembly, and predicted the centromere region according to previously described candidate centromeric tandem repeats[21]. There were 52, 47, and 229 gaps in the Hi-C-based scaffolding assemblies of *S. japonica*, *S. yunnanensis*, and *S. cepharantha*, respectively. After gap filling, only one gap remained in the *S. japonica* genome (4 in *S. yunnanensis* and 11 in *S. cepharantha*). Telomeres and centromeres in *S. japonica* genome were annotated, but centromeres in *S. cepharantha* and *S. yunnanensis* genomes remained unresolved. In *S. yunnanensis* genome assembly, 15 of 26 telomeres were resolved, while

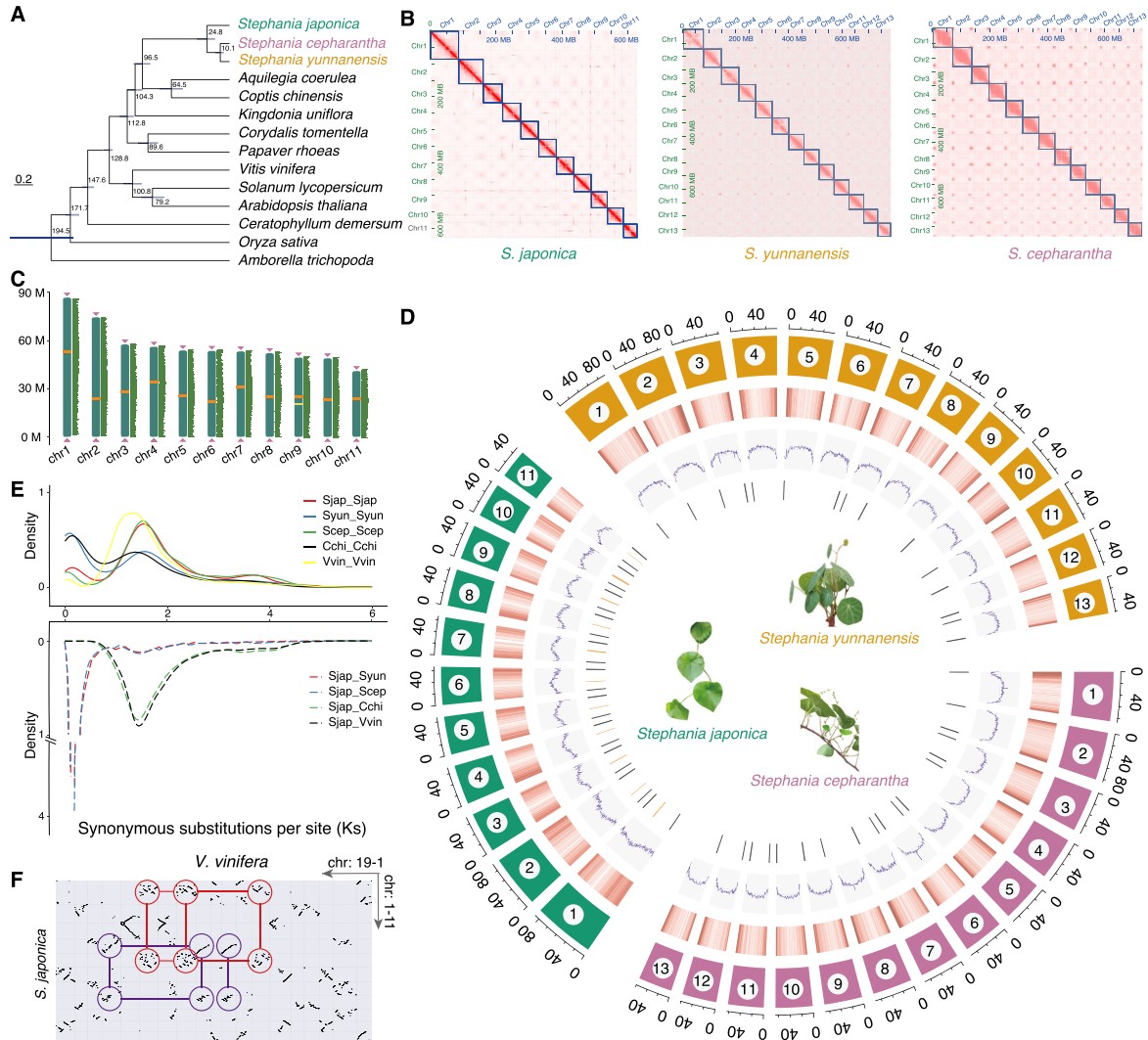

**Fig. 1 | Genome characterization of three *Stephania* species. A** Inferred phylogenetic tree and divergence time estimation of 14 flowering plant species including three *Stephania* species, namely, *Stephania japonica*, *Stephania yunnanensis*, and *Stephania cepharantha*. **B** Genome-wide all-against-all Hi-C interaction map of the three *Stephania* species. **C** Near telomere-to-telomere assembly of *S. japonica*. Orange bars indicate centromere locations, the yellow bar indicates a gap in chromosome 9, and violet triangles indicate telomere locations. **D** Circos plots from the periphery toward the center showing chromosome length (Mb), chromosome number, gene density (red columns), repeat percentage (blue violet lines), and telomere and centromere locations (black and orange columns, respectively). **E** Synonymous substitutions per synonymous site ($K_S$) distributions of paralogous genes (upper panel) and orthologous genes (lower panel) among the three *Stephania* genomes, *Coptis chinensis*, and *Vitis vinifera*. **F** Dot plots of orthologs between *S. japonica* and *V. vinifera*. Red and purple circles highlight two examples suggesting a 2:3 syntenic proportion resulting from whole genome duplication or triplication. Source data are provided as a Source Data file.

*S. cepharantha* genome assembly resolved 18 of 26 telomeres, resulting in a near telomere-to-telomere assembly for *S. japonica* genome and chromosome-level assembly of the *S. cepharantha* and *S. yunnanensis* genomes (Fig. 1C, D, Supplementary Fig. 5).

Whole-genome duplication (WGD) analysis revealed a shared WGD event between *Stephania* genomes and *Coptis chinensis*, as indicated by paralogous gene age distribution (Fig. 1E, Supplementary Fig. 6). Syntenic dot plots offered clear evidence of a 2:3 syntenic proportion between *Stephania* species and *Vitis vinifera*, a core eudicot species that experienced ancestral hexaploidization, unlike *Stephania*[22]. Another WGD event in *Stephania* species occurred after the divergence of its ancestor from *V. vinifera* (Fig. 1F, Supplementary Fig. 6).

### Proposal of the putative cepharanthine biosynthesis pathway

To unravel the genes involved in cepharanthine (**1**) biosynthesis, we retrieved a set of orthologous gene protein sequences from NCBI, including enzymes such as NCS, methyltransferases, and cytochrome

P450 (CYP450). Using these protein sequences as queries, we identified 267, 331, and 234 potential enzyme genes in the genomes of *S. japonica*, *S. yunnanensis*, and *S. cepharantha*, respectively. We narrowed down our focus to genes clustered phylogenetically with known functional proteins related to BIA synthesis in other species. This led to the identification of different number of *NCS*, *OMT*, *NMT* and *CYP450* genes from the three species (Fig. 2A and Supplementary Fig. 7). These candidate genes were distributed conservatively within syntenic regions across the three genomes, despite certain chromosomal fusion and fission events involving *S. japonica* and the other two species (Fig. 2B, Supplementary Fig. 8).

Several candidate genes form putative biosynthetic gene clusters (BGCs), which are conserved across *Stephania* genomes. For instance, a cluster in *S. japonica* chromosome 3 (51.68–51.81 Mb) aligns with *S. cepharantha* chromosome 4 (56.78–56.84 Mb, Fig. 2C). This BGC overlaps with a cluster region identified by plantiSMASH[23], encompassing one *CYP80* and three *6OMTs*. The topologically associating

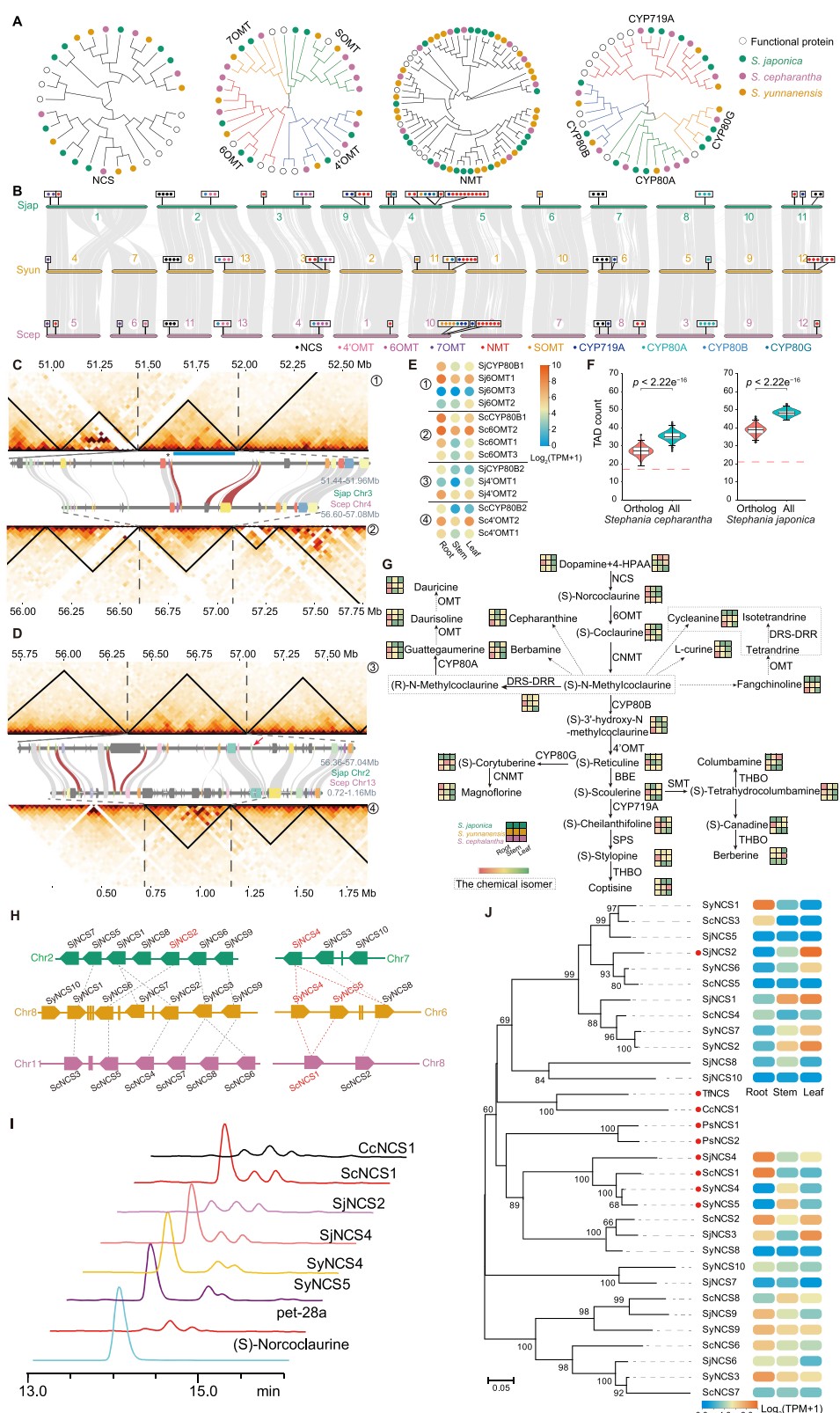

domains (TADs) containing this BGC in both *S. japonica* and *S. cepharantha* closely resemble each other. However, gene expression patterns within the gene cluster in *S. japonica* do not entirely align between the two species. For instance, *SjCYP80B1*, *Sj6OMT1*, their orthologous genes, *ScCYP80B1*, and *Sc6OMT2*, are highly expressed in roots in both species. Yet, *Sj6OMT3* and *Sj6OMT2* are highly expressed in *S. japonica* leaves, whereas their orthologous genes, *Sc6OMT1* and

*Sc6OMT3*, show high expression in roots (Fig. 2E). Another example is a conserved cluster found in *S. japonica* chromosome 2 (56.44–56.62 Mb) and *S. cepharantha* chromosome 13 (0.78–0.86 Mb), including one *CYP80* and two *4'OMTs* (Fig. 2D). This cluster shows a general co-expression pattern. *SjCYP80B2*, *Sj4'OMT2*, along with their orthologous genes, *ScCYP80B2* and *Sc4'OMT1*, are highly expressed in roots. Similarly, *Sj4'OMT1* and its orthologous gene *Sc4'OMT2* exhibit

**Fig. 2 | Identification of putative enzymes involved in cepharanthine biosynthesis. A** Phylogenetic trees of known enzyme genes and candidate *Stephania* genes, including norclaurine synthase (NCS), *O*-methyltransferase (OMT), *N*-methyltransferase (NMT), and cytochrome P450 (CYP450). **B** Collinearity among the three *Stephania* species, namely, *Stephania japonica* (Sjap), *Stephania yunnanensis* (Syun), and *Stephania cepharantha* (Scep). Gray lines represent synteny patterns between the genomic regions of neighboring *Stephania* genomes. Locations of key enzymes in the cepharanthine biosynthetic pathway, namely, NCS, sterol-methyltransferase (SMT), OMT, NMT, and CYP450, are highlighted with colored points. **C** Illustration of a putative biosynthetic gene cluster (BGC) located in *S. japonica* chromosome 3 and *S. cepharantha* chromosome 4. The topologically associating domains (TADs) present in this cluster are conserved between these two species. The blue band with a star denotes the location of a BGC identified using plantiSMASH. Synteny of four biosynthetic genes is highlighted with dark red links. **D** Illustration of a putative BGC located in *S. japonica* chromosome 2 and *S. cepharantha* chromosome 13. The red arrow denotes the position at which the boundary of the *S. cepharantha* TAD projected onto its corresponding *S. japonica*

TAD. **E** Expression of BGC genes. Circled numbers denote their locations in Panels C and D. **F** Enrichment of candidate biosynthetic genes in the same TAD. Violin plots depict TAD count distributions of 1,000 permutations sampled from orthologs or all genes. Dashed red lines denote TAD counts of candidate genes, that is, 17 in *S. japonica* and 21 in *S. cepharantha* (box plot represents median and 25th and 75th percentiles-interquartile range; IQR-and whiskers extend to maximum and minimum values; *n* = 1000 permutations; statistical analysis: two-sided Wilcox test). **G** Proposed biosynthetic pathway for BIA in *Stephania* species. Each cube represents the concentration of its closest compound in different tissues, namely, the root, stem, and leaf, of three *Stephania* species. **H** Genomic location of two syntenic regions. *NCS* genes were denoted as irregular pentagons, whereas other genes were denoted as vertical lines. Ortholog gene pairs were linked with dotted lines. **I** Product profiles of NCS using **4** and **5** as substrates. CcNCS1, *NCS* gene of *Coptis chinensis* whose enzymatic activity is known. The experiment has been repeated three times with similar results. **J** Phylogenetic relationship and expression of *NCS* genes. Bootstrap values (>50) were labeled in the corresponding branches. Source data are provided as a Source Data file.

high expression in leaves (Fig. 2E). We explored whether these candidate BIA synthetic genes tend to reside within the same TAD. In *S. japonica*, 50 candidate genes were present in 21 TADs. To ascertain significance, we compared two background gene sets: one from 1000 permutations of the 267 orthologous genes and the other involving all genes. The former set occupied fewer TADs than the genome-wide set (median number of TADs: 27 vs. 36, *p* < 2.22e$^{-16}$, two-sided Wilcox test). Additionally, candidate genes were present in fewer TADs than both background sets (empirical *p*-value < 0.001, Fig. 2F). This pattern is repeated in *S. cepharantha*. Co-regulation and co-inheritance are two hypotheses for BGC function[24]. Taken together, the co-occurrence of BGCs in the conserved syntenic TADs of two species that diverged nearly 25 Mya is in line with the co-inheritance hypothesis. At the same time, our data partially support the co-regulation hypothesis. When comparing the BGCs between *S. cepharantha* vs. *S. yunnanensis* and *S. japonica* vs. *S. yunnanensis*, the same pattern could be observed (Supplementary Fig. 9). The enrichment of putative biosynthetic genes in the TAD suggests that these genes are likely to be functionally related, indicating that they are involved in the same *Stephania*-specific BIA biosynthetic pathway.

Based on the biosynthetic gene mining, we employed untargeted metabolomics to profile BIAs across various tissues of the three *Stephania* species (Fig. 2G, Supplementary Figs. 10 and 11). Prior *Stephania* species studies identified several BIAs such as coclaurine (**11**), reticuline (**12**), cepharanthine (**1**), and eight others in *Stephania hainanensis*[25], 18 BIAs in *Stephania kwangsiensis*[26], and 13 BIAs across five *Stephania* species[27]. Using untargeted metabolite profiling, we detected two BIA precursors, **4** and **5**, alongside 23 BIA-type structures. The (*S*) and (*R*) configurations of *N*-methycoclaurine (**7**, **13**), integral to BIA dimer formation, could not be discerned, but both configurations should exist in *Stephania* species. Similarly, tetrandrine (**3**), isotetrandrine (**14**), and cycleanine (**15**), being chemical isoforms, were indistinguishable. These compounds exhibited predominant accumulation in roots, followed by leaves and stems. While the biosynthesis of BIA monomers like berberine (**2**) and coptisine (**16**) is well-established, the mechanisms underlying BIA dimer biosynthesis remain to be elucidated (Fig. 2G).

We subsequently validated the putative genes for BIA biosynthesis. All the *NCS* members from all three species were evaluated, particularly those within two syntenic regions (Fig. 2B, H). These regions encompassed multiple several *NCS* homologs, forming complete or split gene pairs and arrays[24]. To avoid overlooking possible *NCS* genes, we conducted local searches for neighboring genes (blast search with E-value = 1e$^{-10}$). The first region comprised seven, seven, and six *NCS* copies in *S. japonica*, *S. yunnanensis*, and *S. cepharantha*, respectively, whereas the second region contained three, three, and two *NCS* copies in these three respective genomes. Catalytic experiments involving 12

NCSs (11 *Stephania NCSs* with high expression and one known-function NCS from *Coptis chinensis* as a positive control) were conducted using 4 and 5 as substrates (Fig. 2I). The results revealed that five candidate *Stephania NCS* genes possessed NCS functionality. Four out of these five *Stephania NCSs* exhibited superior enzymatic activities compared with the *Coptis chinensis* NCS. These four *NCSs* were all situated in the second syntenic region (Fig. 2H, I). Phylogenetic analysis unveiled that these four *Stephania NCSs* formed a monophyletic group, with two *S. yunnanensis NCSs* emerging as paralogs due to a recent duplication event (Fig. 2J). These two *S. yunnanensis NCSs* were highly expressed in stems, whereas *ScNCS1* and *SjNCS4* exhibited high expression in roots. These results underscore the dynamic evolution of *Stephania NCS* genes in terms of both local copy number adjustments and changes in expression patterns.

## Broad spectrum anti-coronavirus activity of cepharanthine analogs

We assessed the antiviral potential of 23 compounds from the cepharanthine (**1**) biosynthetic pathway, which encompassed dopamine, 12 BIA monomers, and 10 cepharanthine (**1**) analogs (Fig. 3, Supplementary Fig. 12, Supplementary Data 1). This comprehensive investigation involved SARS-CoV-2 (Table 2) and three SARS-CoV-2-related viruses: Guangxi pangolin-CoV (GX_P2V)[28,29] (Fig. 3A, Supplementary Fig. 12), SARS-CoV-2 virus-like particles (SARS-CoV-2 trVLPs[30], Fig. 3B), and vesicular stomatitis virus (VSV) pseudotyped virus (SARS-CoV-2 pseudotyped virus[31], Fig. 3C). GX_P2V shares a high amino acid similarity of 92.5% in the spike (S) protein with SARS-CoV-2, utilizes the same receptor angiotensin-converting enzyme 2 (ACE2), and does not infect humans. The SARS-CoV-2 trVLP system swaps the nucleocapsid (N) protein-coding sequence of SARS-CoV-2 with GFP, generating SARS-CoV-2-GFP/ΔN virions, which complete their viral life cycle only with exogenous SARS-CoV-2 N protein supplementation. The SARS-CoV-2 pseudotyped virus expresses the full-length S protein of SARS-CoV-2. All three viruses are amenable to study in biosafety level 2 (BSL-2) laboratories.

In our initial assessment, antiviral activity against GX_P2V were first performed (Fig. 3A, Supplementary Fig. 12). Tetrandrine (**3**) was the only compound exhibiting detectable cytotoxicity (half-cytotoxic concentration = 21.9 μM). Eight compounds displayed dose-dependent inhibitory effects, with half-maximal effective concentrations (EC50) ranging from 1.49 to 15.73 μM. All these compounds were BIA dimers. While cepharanthine (**1**) exhibited satisfactory inhibition (EC50 = 1.78 μM), cycleanine (**15**) performed superior to cepharanthine (**1**) (EC50 = 1.49 μM). The anti-SARS-CoV-2 activities of these seven compounds were further confirmed using live SARS-CoV-2 virus (SARS-CoV-2/WH-09/human/2020/CHN; GenBank: MT093631.2) in biosafety level 3 lab. Seven compounds inhibited SARS-CoV-2 infection

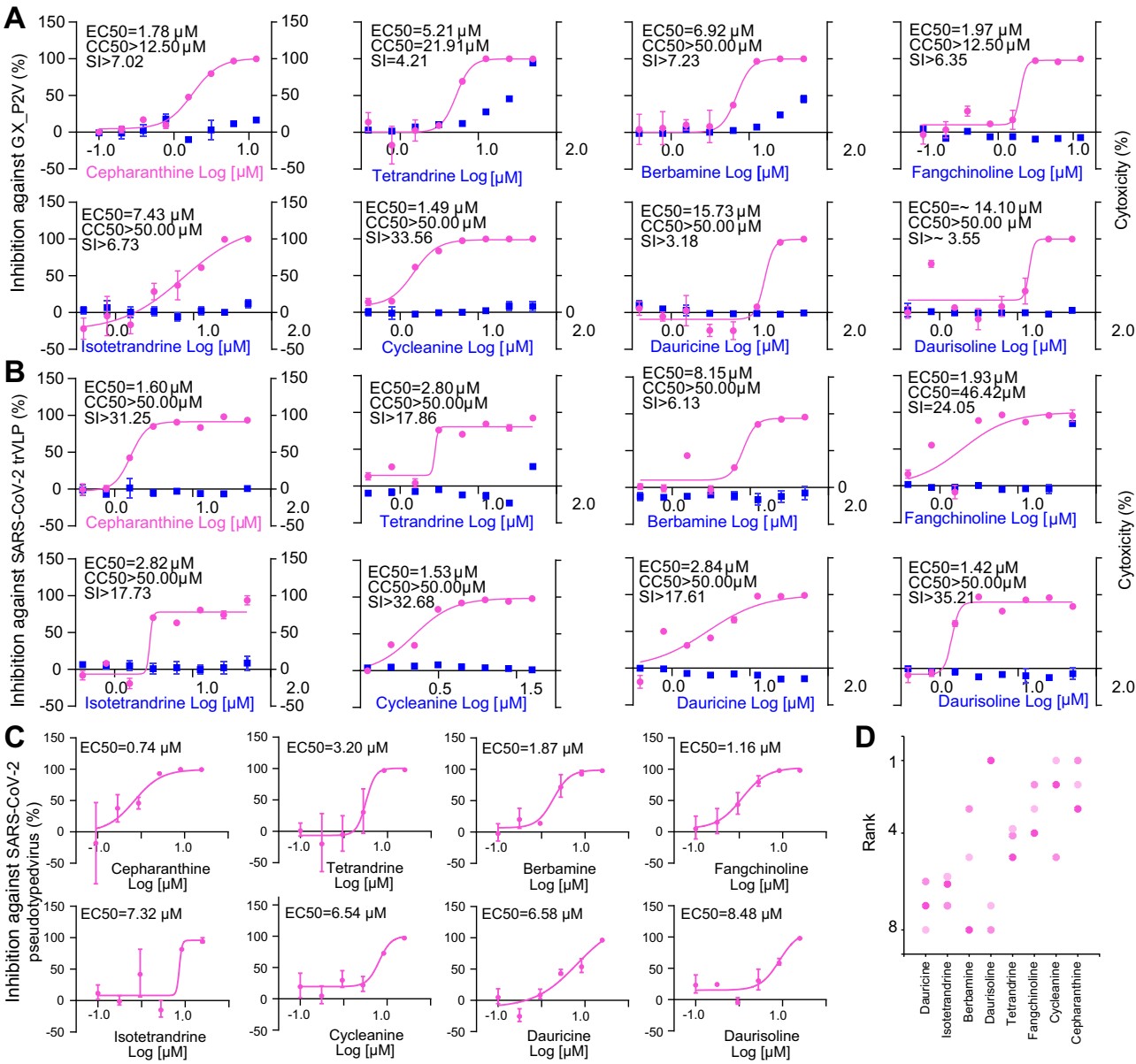

**Fig. 3 | Anti-GX_P2V, SARS-CoV-2 and SARS-CoV-2 pseudovirus activity assays of eight BIAs.** Vero E6 cells were infected with GX_P2V (MOI = 0.01) and Caco 2-N cells were infected with SARS-CoV-2 trVLP (MOI = 0.01), and both cells were treated with different concentrations of BIAs. Intracellular viral loads were detected at 48 hpi using quantitative reverse transcription–polymerase chain reaction assay and normalized with *GAPDH* in Vero E6 cells (**A**) and Caco 2-N cells (**B**). Cytotoxicity of these BIAs to Vero E6 cells (**A**) and Caco 2-N cells (**B**) was measured according to the CellTiter-Blue® assay. The y-axis on the left and right graphs represent mean percentage of inhibition against virus yield and cytotoxicity of BIAs, respectively. **C** Inhibitory analysis of SARS-CoV-2 pseudovirus infection to BHK21-ACE2 cells by different doses of BIAs. The experiment has been repeated three times with similar results. **D** Ranks of eight BIAs in anti-SARS-CoV-2 assays. Colors represent three assays. The data in panels A-C are presented as mean values +/– s.d. (*n* = 3 biologically independent samples). Source data are provided as a Source Data file.

in a dose-dependent manner (and dauricine (**18**) can inhibit SARS-CoV-2 wild type, delta, omicron strains with an EC50 range of 18.2 to 33.3 μM in the Vero E6 cells[32]), with EC50 ranging from 2.19 to 19.95 μM. Among them, cepharanthine (**1**) was also the most effective compound against SARS-CoV-2, with an EC50 of 2.19 μM, which was consistent with that of remdesivir (**30**). Subsequently, we explored the antiviral activity of these eight compounds against SARS-CoV-2 trVLPs. Each of the eight compounds displayed dose-dependent inhibitory effects, with EC50 values ranging from 1.42 to 8.15 μM. Additionally, none of these compounds demonstrated cytotoxicity. Both cycleanine (**15**) and cepharanthine (**1**) maintained consistent efficacy across both GX_P2V- and SARS-CoV-2 trVLP-based assays, while daurisoline (**17**) (EC50 = 1.42 μM) exhibited significant potency against SARS-CoV-2 trVLPs but demonstrated a considerably higher EC50 value against

GX_P2V (EC50 = 14.1 μM). Lastly, we investigated the antiviral activity of these eight compounds against the SARS-CoV-2 pseudotyped virus. Cepharanthine (**1**) exhibited the highest effectiveness in this assay (EC50 = 0.74 μM), trailed by fangchinoline (**19**) (EC50 = 1.16 μM) and berbamine (**10**) (EC50 = 1.87 μM). To recap, all eight BIA dimers consistently displayed efficacy against diverse SARS-CoV-2-related viruses, with EC50 values consistently below 20 μM. Among these, cepharanthine (**1**) and fangchinoline (**19**) both exhibited EC50 values consistently below 2.0 μM. Compound 15 outperformed cepharanthine (**1**) in GX_P2V- and SARS-CoV-2 trVLP-based assays and ranking as the second-best compound in terms of all three assessments (Fig. 3D). We employed remdesivir (**30**), an approved anti-SARS-CoV-2 compound, as a positive control to assess its antiviral activity against GX_P2V and SARS-CoV-2 trVLP (Supplementary Fig. 13). **30** displayed an EC50 value

**Table 2 | EC50 assay of BIAs and remdesivir against SARS-CoV-2 live virus**

| Compound | SARS-CoV-2 (Vero E6) EC50 (μM) |
|---|---|
| Cepharanthine (1) | 2.19 |
| Tetrandrine (3) | 3.13 |
| Berbamine (10) | 19.95 |
| Fangchinoline (19) | 5.01 |
| Isotetrandrine (14) | 4.47 |
| Cycleanine (15) | 8.91 |
| Daurisoline (17) | 12.50 |
| Remdesivir (30) | 2.19 |

of 1.64 μM against GX_P2V, which was intermediate between 1 and 15. For SARS-CoV-2 trVLPs, 30 exhibited an EC50 of 0.11 μM, superior to all tested BIAs. To discern at which stage of the GX_P2V or SARS-CoV-2 trVLP life cycle these compounds exerted their effects, we evaluated their antiviral activity through time-of-addition assays within a single infectious cycle (Supplementary Fig. 14). Dauricine (18) displayed superior inhibitory activity during the entry stage compared with the post-entry stage. In contrast, 1 and 19 exhibited similar inhibitory effects in both entry and post-entry stages, while other compounds generally demonstrated enhanced efficacy in the post-entry stage.

We extended our exploration to investigate the broad-spectrum anti-coronavirus activity of these eight BIAs (1, 3, 10, 19, 14, 15, 18, and 17) against multiple coronaviruses (Fig. 4A–D). These assays were performed in four coronaviruses: PEDV (Fig. 4A, $EC_{50}$ = 0.20, 0.44, 0.50, 0.20, 1.67, 0.34, 1.70, and 0.82 μM, respectively), SADS-CoV (Fig. 4B, $EC_{50}$ = 1.03, 2.19, 5.84, 2.23, 3.67, 2.77, 0.84, and 2.80 μM, respectively), severe acute respiratory syndrome coronavirus (SARS-CoV) pseudotyped virus (Fig. 4C, $EC_{50}$ = 1.37, 3.04, 5.55, 1.73, 4.77, 3.2, 2.63, and 2.76 μM, respectively), and Middle East respiratory syndrome coronavirus (MERS-CoV) pseudotyped virus (Fig. 4D, $EC_{50}$ = 2.19, 2.17, 3.37, 4.47, 3.31, 2.45, 2.13, and 2.46 μM, respectively). These eight BIAs demonstrated generally better inhibitory effects against these coronaviruses than against SARS-CoV-2; an overall comparison between other anti-coronavirus assays and the anti-SARS-CoV-2 assays indicated that the former group exhibited more potent antiviral activity than the latter one ($p$ = 0.03, two-sided t-test). Compound 18 outperformed 1 in its efficacy against PEDV and MERS-CoV (Fig. 4E). Across all seven coronaviruses, cepharanthine (1) consistently demonstrated robust performance, consistently ranking among the top three (Fig. 4F). 17, 18, and 15 occasionally secured the first rank. These findings underscore the potential pan-coronavirus inhibitory activity of BIA dimers.

## Antiviral activity is related to structural changes among *Stephania* BIA dimers

Our extensive dose-response analysis unveiled that of the 23 compounds assessed along the pathway, eight BIA dimers, including cepharanthine (1), exhibit bioactivity, whereas the BIA precursor 4, two BIA dimers, and 12 BIA monomers do not. This unexpected discrepancy (8/10 vs. 0/13; $p$ = 0.005, one-sided Fisher's exact test) in activity between the tested dimers and monomers suggests the presence of mechanistic explanations, both biological and chemical, rather than a result of random variation. Differences in antiviral activity along the pathway imply that the activity may arise from two types of structural changes: one related to BIA dimerization and another related to the conformations, configurations, and modifications of BIA dimers.

The biosynthesis of BIA dimers, unlike BIA monomers, remains less documented. While many natural BIA dimers form from the coupling of (R)-N-methylcoclaurine (13) and (S)-N-methylcoclaurine

(7)-such as magnoline (31), berbamunine (9), and guatteguamerine (32)[18], only 32 was identified in *Stephania* by metabolomics screening, yet it lacked antiviral activity (Fig. 5A). By contrast, daurisoline (20), potentially an OMT product derived from 32, demonstrated potent anti-SARS-CoV-2 activity. Consequently, our results did not fully support the assumption that dimerization is the pivotal step for acquiring antiviral function. Compounds such as 10, 15, 19, and cepharanthine (1) are likely formed through dimerization catalyzed by CYP80A, coupled with methylation by OMT.

A systematic survey of intermediate compounds in cepharanthine (1) biosynthesis pathway may reveal the crucial reactions and enzymes for producing BIA dimers with anti-coronavirus ability. Compound 3, 14, and 15 represent conformational and configurational isomers. Variations in their antiviral activity and cytotoxicity imply the impact of structural changes in isomers on pharmaceutical properties. Within the group of BIA dimers, no consistent trend of increasing efficacy in products compared to their precursors emerged. For example, 3 and 14, OMT products derived from fangchinoline (19)-demonstrated weaker antiviral activity compared to their precursor. This observation contradicts the null hypothesis that antiviral activity strengthens along the biosynthetic pathway, emphasizing the need to explore and delineate as many compounds as possible within the pathway. Molecular docking calculations provided similar results (Fig. 5B, C). The binding energy between BIA dimers and the SARS-CoV-2 spike protein-ACE2 complex (7VX4 or 7VX5) was significantly higher than that of BIA monomers ($p$ = 6.20e$^{-0.5}$/7.92e$^{-0.5}$, two-sided Wilcoxon test, Fig. 5B). However, no clear correlation existed between antiviral activity and binding energy of BIA dimers (Fig. 5C). Furthermore, no evidence demonstrated that the conformation or modification of BIA dimers influenced their antiviral activity.

Elucidating a well-defined mechanism of action for BIAs against SARS-CoV-2 is crucial for understanding the relationship between structure and antiviral activity, and accelerating drug discovery. The inhibitory effects of bis-BIAs on SARS-CoV-2 and other coronaviruses have been attributed to various mechanisms[33–35], with the blockade of calcium ion ($Ca^{2+}$)-dependent membrane fusion by BIAs emerging as a plausible explanation. Bis-BIAs are known as hindering calcium inward flow through the physical alteration of lipid properties[36], thereby suppressing calcium-induced responses and inhibiting membrane permeability transition (MPT)[37,38]. This cascade results in inhibiting $Ca^{2+}$-mediated fusion and suppressing virus entry[34]. Variations in the conformation, configuration, and modification of BIA dimers could feasibly impact the physical alteration of lipid properties at varying levels, thereby influencing their antiviral activity.

## Other genera with antiviral compound discovery potential

Considering that *Stephania* yields a cluster of antiviral compounds, we were intrigued to determine whether the antiviral activity was unique to the *Stephania* clade or extended to a broader range of plants producing bioactive cepharanthine (1) analogs. BIAs are largely restricted to certain plant families primarily in the order Ranunculales[39]. Consequently, BIA metabolism research has mainly focused on plants from the families *Papaveraceae*, *Menispermaceae*, *Ranunculaceae*, and *Berberidaceae*[40–42] Among them, cepharanthine (1) and tetradrine (3), derived from the *Menispermaceae* family, have demonstrated anti-SARS-CoV-2 activity. Additionally, BIAs are sporadically found in the orders Piperales and Magnoliales as well as in the families *Rutaceae*, *Lauraceae*, *Cornaceae*, and *Nelumbonaceae*[43]. To date, BIAs have not been identified in gymnosperms, monocots, or Amborellales. We performed a search to identify eudicot plants producing BIAs with reported potential antiviral activity. Neferine (33) and liensinine (34), major bisbenzylisoquinoline components and bioactive constituents of sacred lotus (*Nelumbo nucifera*)[44], have shown potential as SARS-CoV-2 inhibitors[45,46]. Sacred lotus, a basal eudicot plant, produces numerous BIAs[47,48]. For our study, we selected the genus *Thalictrum*

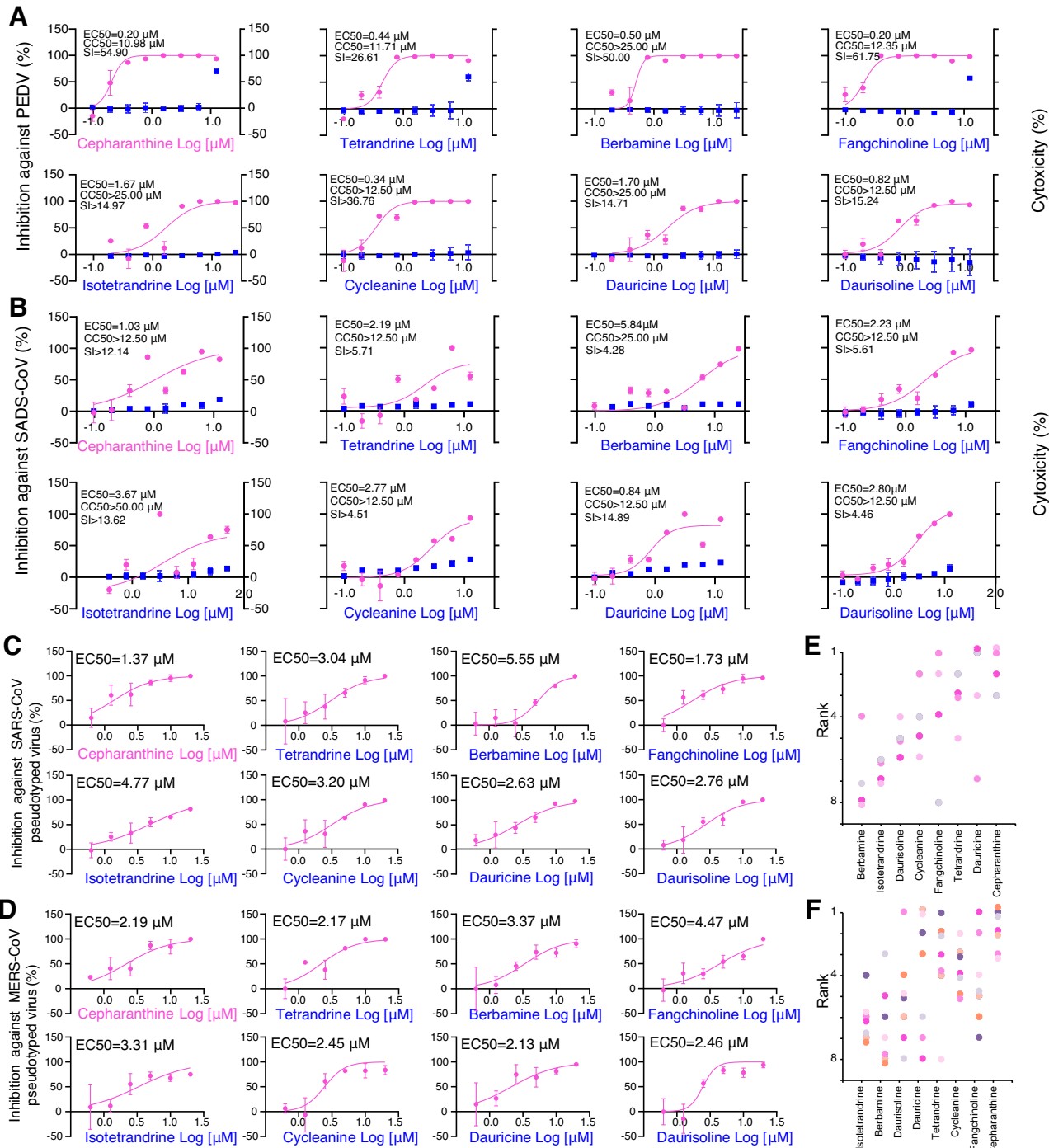

**Fig. 4 | Broad spectrum anti-coronavirus activity assessment of eight BIAs.** Huh 7 cells were infected with PEDV (MOI = 0.01) and SADS-CoV (MOI = 0.01) and treated with different concentrations of BIAs. Intracellular loads of PEDV (**A**) and SADS-CoV (**B**) were detected at 48 hpi by qRT-PCR and normalized by *GAPDH*. Cytotoxicity of these drugs to Huh 7 cells was measured according to the CellTiter-Blue® assay. The y-axis on the left and right graphs represents mean percentage of inhibition against virus yield and cytotoxicity of BIAs, respectively. Inhibitory analysis of SARS-CoV pseudovirus infection on BHK21-ACE2 cells (**C**) and MERS-CoV pseudovirus infection on Huh 7 cells (**D**) by different doses of BIAs. Magenta points and lines denote the antiviral activity of the compounds against four anti-coronavirus, respectively. The blue box represents cytotoxicity of BIAs. The experiment has been repeated three times with similar results. **E** Ranks of eight BIAs in four anti-coronavirus assays, including PEDV, SADS, SARS-CoV pseudotyped virus, and MERS-CoV pseudotyped virus. **F** Ranks of eight BIAs in all seven anti-coronavirus assays. The data in panels A-D are presented as mean values +/− s.d. (n = 3 biologically independent samples). Source data are provided as a Source Data file.

among Ranunculales plants, considering its taxonomic proximity to BIA-producing plants[49]. Our search yielded five antiviral analogs from *Nelumbo* and *Thalictrum* (Fig. 6A): **33** and **34** from *Nelumbo*, and thalidezine (**35**), thalmineline (**36**), and methoxyadiantifoline (**37**) from

*Thalictrum*. No BIAs have been reported in gymnosperms. Moreover, Amborellales, early-diverging angiosperms, do not accumulate these compounds[50]. Consequently, BIA biosynthesis appears to have evolved after the divergence of Amborellales and other angiosperms but

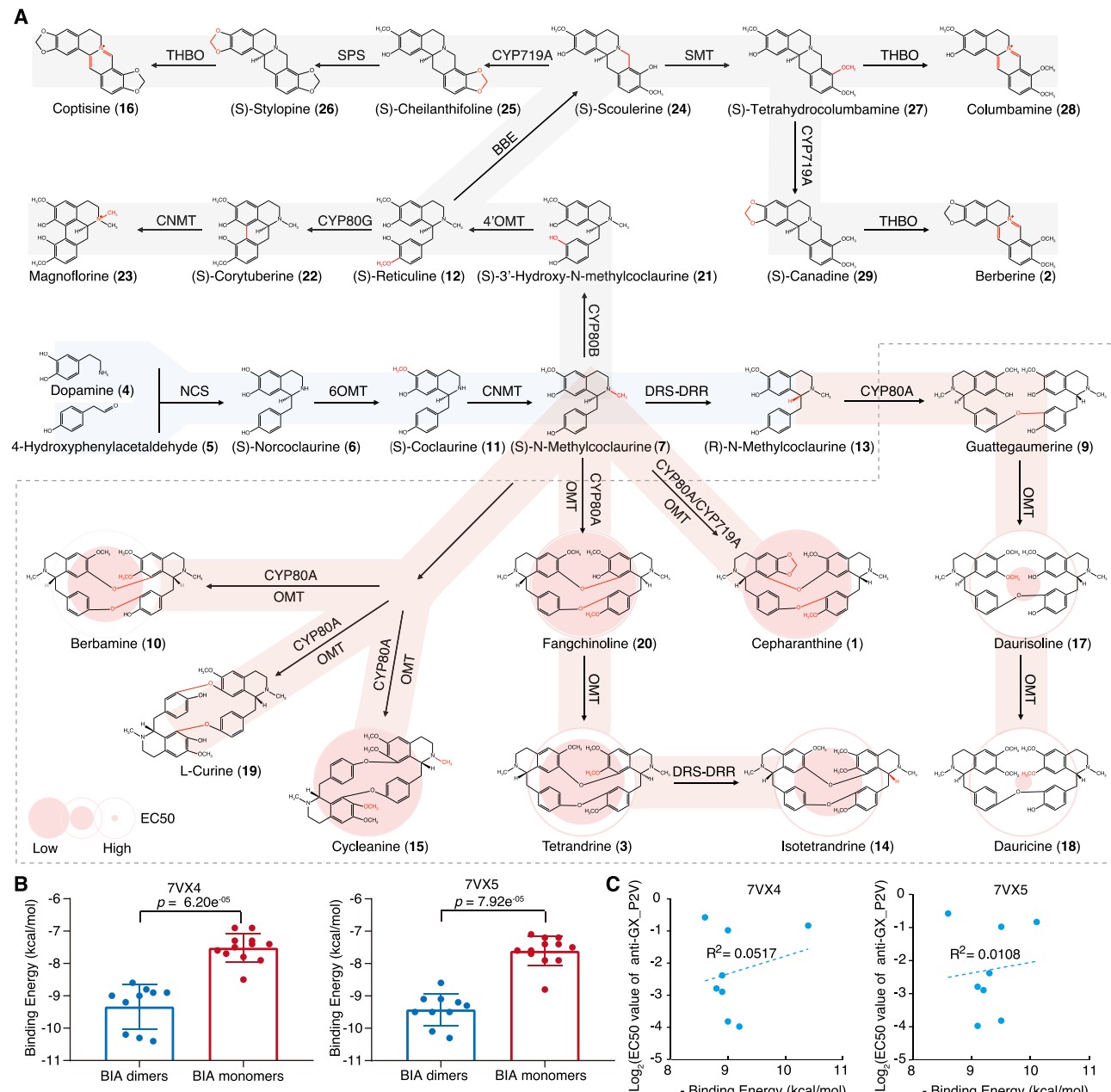

**Fig. 5 | Antiviral activity of BIA predominantly occurs in the dimer branches. A** Two precursors and four other backbone BIA compounds; twelve other BIA monomers; and BIA dimers, including cepharanthine (**1**), are labeled with light blue, light gray, and light pink backgrounds, respectively. **1** and other dimers are surrounded by gray dashed lines. The anti-GX_P2V activities of cepharanthine (**1**) and seven dimers are represented by two-layer circles, where the diameter of the inner layer is negatively proportional to the compound's EC50 value. Dopamine (**4**), BIA monomers, and dimers without antiviral activity are represented as smaller hollow circles. **B** Molecular docking of *Stephania* BIAs with SARS-CoV-2 spike protein-ACE2 complex. The left panel summarizes the binding pattern of BIAs to 7VX4 (ACE2-RBD in SARS-CoV-2 Beta variant S-ACE2 complex), while the right panel summarizes the binding pattern of BIAs to 7VX5 (ACE2-RBD in SARS-CoV-2 Kappa variant S-ACE2 complex). The data are presented as mean values +/− s.d. (statistical analysis: two-sided Wilcox test). **C** Correlation of EC50 values and binding energy of antiviral BIA dimmers. The X-axis is the original value times minus one. Source data are provided as a Source Data file.

before the separation of magnoliids, eudicots, and monocots (Fig. 6B). Moreover, certain compounds with anti-SARS-CoV-2 activity originate from several early-diverging eudicots such as *Nelumbo* from Proteales and *Thalictrum* and *Stephania* from Ranunculales. Two explanations may account for the phylogenetic distribution of plants that produce and accumulate antiviral cepharanthine (**1**) analogs: first, the activity being evolved in three lineages independently, and second, the activity emerging in the early history of angiosperm evolution but being lost subsequently in some lineages.

## Discussion

The concept of structure-activity relationship, where structurally similar compounds are expected to share similar pharmaceutical functions, has been employed to identify effective antibacterial drugs[51]. Our findings indicate that, similar to cepharanthine (**1**), a group of cepharanthine (**1**) analogs possess potential broad-spectrum anti-coronavirus activity. Furthermore, it is crucial to underscore that this group of analogs stems from the same plant clade as that of cepharanthine (**1**). The chemodiversity of natural BIAs evolved over millions

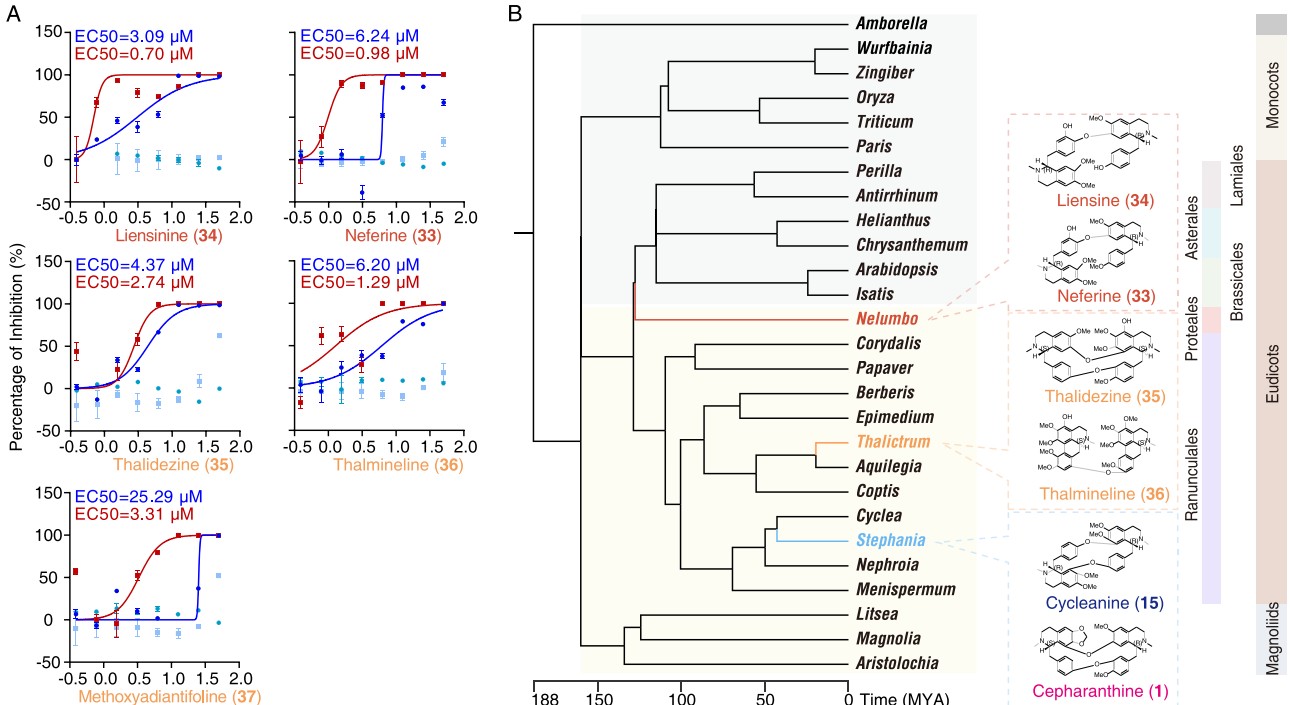

**Fig. 6 | *Stephania*, *Thalictrum*, and *Nelumbo* as the origin plants for antiviral cepharanthine analogs. A** Antiviral activity and cytotoxicity of cepharanthine (**1**) analogs originating from other plants. Dark red and blue points and lines denote the antiviral activity of the compounds against GX_P2V and SARS-CoV-2 trVLPs, respectively. Dark red and blue points denote their cytotoxicity. The data are presented as mean values +/− s.d. (*n* = 3 biologically independent samples). **B** Phylogenetic positions of three plants that could produce antiviral cepharanthine (**1**) analogs. Angiosperm plants, namely, magnoliids, Ranunculales, and Proteales, capable of producing benzylisoquinoline alkaloids are labeled with a light-yellow background, whereas other angiosperm plants, namely, Amborellales, monocots, and core eudicots, are labeled with a light gray background. Data of the phylogeny and divergence time were retrieved from TimeTree (http://www.timetree.org). Source data are provided as a Source Data file.

of years since the split between early-diverging angiosperms. During that time, plants continuously experienced environmental challenges including pathogens and could gradually and successfully adapt to such pressures through the production of bioactive metabolites within an extensive chemical space[52,53], thus responding to environmental challenges. This evolution engendered a group of compounds that engaged in an arms race between plants and their associated organisms, such as pathogens, insects, and herbivores[54,55]. We hypothesized that understanding the genetic basis and evolutionary trajectory underlying the biosynthesis of active compounds during the plant-vs.-pathogen arms race would greatly benefit efficient drug discovery, mirroring the human-vs.-pathogen arms race.

As potentially promising anti-coronavirus agents, the commercial production of cepharanthine (**1**) and other *Stephania*-derived BIAs such as **3** and **15**, could present significant supply challenges because of their heavy reliance on natural plant extracts, similar to the taxol supply crisis case[56]. Our high-quality genome assembly and annotation provide valuable resources for environmentally sustainable green production through synthetic biology or breeding. Through a comprehensive evaluation of antiviral activity across the cepharanthine (**1**) biosynthetic pathway, we identified cepharanthine (**1**) analogs, such as **15** and **19**, that exhibit comparable potency to cepharanthine (**1**).

Prior research has extensively elucidated biosynthetic genes associated with BIA monomers, like berberine (**2**), sanguinarine (**38**), and morphine (**39**), but knowledge about BIA dimer biosynthesis remains limited[11]. In this study, we first assemble high-quality *Stephania* genomes, and then examine the anti-coronavirus activity of cepharanthine (**1**) analogs and BIAs from a biosynthetic pathway perspective. We hope these findings will expedite the search for broad-spectrum anti-coronavirus. The inception of BIAs likely dates back to early angiosperms, specifically after the split between Amborellales

and Nympheales/Austrobaileyales (Fig. 6). The Ranunculales genome sequences provide additional insights into the evolution of early-diverging eudicot lineages and their chemodiversity[19]. Ultimately, we advocate for increased focus on the biological facet of natural product research[57]. Integrating herbgenomics and green production techniques fortify the groundwork for future pharmaceutical applications of natural products.

## Methods

### Plant materials, DNA/RNA library construction, and sequencing

Fresh leaves of *Stephania japonica*, *Stephania yunnanensis*, and *Stephania cepharantha* were collected in Yunnan and Hubei Provinces, China. High-quality genomic DNA was isolated from the fresh leaves using the hexadecyl trimethyl ammonium bromide method[58]. The quality and concentration of the isolated genomic DNA were assessed through 0.75% agarose gel electrophoresis, NanoDrop™ One spectrophotometry (NanoDrop Technologies, Wilmington, DE, USA), and Qubit 3.0 Fluorometry (Life Technologies, Carlsbad, CA, USA).

The genomic DNA was randomly sheared using a Covaris ultrasonic disruptor. Pair-end libraries with an insert size of ~300 bp were prepared using the VAHTS Universal DNA Library Prep Kit for MGI (Vazyme, Nanjing, China). Sequencing was performed using the MGI DNBSEQ-T7 platform (Beijing Genomics Institute, Shenzhen, China). Raw reads were cleaned to discard low-quality reads (i.e., reads with adaptors, unknown nucleotides, or more than 20% low-quality bases) using SOAPnuke (v2.1.4, https://github.com/BGI-flexlab/SOAPnuke). The clean data were used for subsequent analyses.

For Oxford Nanopore ultralong sequencing (reads N50 > 50 kb), the libraries were prepared using the SQK-ULK001 Ultra-long DNA Sequencing Kit. The purified library was loaded onto primed R9.4 Spot-On Flow Cells and sequenced using the PromethION sequencer

(Oxford Nanopore Technologies, Oxford, UK) with 48-h runs at Wuhan Benagen Tech Solutions Company Limited (Wuhan, China). Base-calling analysis of raw data was performed using Oxford Nanopore GUPPY (version 0.3.0).

Total RNA was extracted from nine samples-three types of tissues (root, stem, and leaf) of three species-using the RNA prep Pure Plant Plus Kit (Tiangen Biotech Co., Ltd., Beijing, China). RNA samples were pooled, and a strand-switching method and the cDNA–polymerase chain reaction (PCR) Sequencing Kit (SQK-PCS109) were used to perform cDNA sequencing on a PromethION sequencer (Oxford Nanopore Technologies). Low-quality raw sequencing reads were filtered to obtain clean data and used for subsequent analysis.

High-quality DNA extracted from young leaves of the three species was used for Hi-C sequencing. Formaldehyde was used for fixing chromatin. In situ Hi-C chromosome conformation capture was performed according to the DNase-based protocol[59]. The libraries were sequenced using the 150-bp paired-end mode on the MGI DNBSEQ-T7 platform.

### Genome assembly and pseudochromosome construction

Based on BGI short-read sequencing data of the three species, $k$-mer analysis with a $k$ value of 19 was performed to estimate the genome size and heterozygosity using the kmer_freq program in the GCE package (v.1.0.0, option: -H 1). Sequencing depth was estimated by determining the highest peak value in the frequency curve of the $k$-mer occurrence distribution. Genome sequences of *S. japonica*, *S. yunnanensis*, and *S. cepharantha* were obtained using Oxford Nanopore ultralong sequences. Genomic assembly was performed using NextDenovo (options: read_cutoff = 1 k, blocksize = 1g, parallel_jobs = 8; sort options: −m 12 g −t 8 −k 40; minimap2 options: -x ava-ont −t 8 −k17 -w17; nextgraph options: -a1, https://github.com/Nextomics/Nextdenovo). Error correction was performed for two rounds on the assembly results based on the nanopore and DNBSEQ-T7 sequencing data using Racon (v1.4.11, options: -t 30, https://github.com/isovic/racon) and Pilon (v1.23, options: --changes --diploid --threads 30 --fix all), respectively.

For pseudochromosome-level scaffolding, ALLHIC (v0.9.12) was used for stitching, after which we imported the final files (.hic and.assembly) generated by ALLHIC into Juicebox (v1.11.08, https://github.com/aidenlab/Juicebox) for manual optimization. We filled gaps in the pseudomolecules using Winnowmap (V1.11, https://github.com/marbl/Winnowmap, options: k = 15, --MD). Finally, redundant heterozygous sequences of the genome were removed using the Purge_haplotigs pipeline (v.1.0.4, https://github.com/skingan/purge_haplotigs_multiBAM, option: purge -a 60) to obtain the final assembly.

### Identification of the telomere and centromere regions

For telomere identification, all ONT long reads that overlapped with the 50-bp flanking region of the start or end base of a chromosome were collected. Telomeric sequences (CCCTAAA/TTTAGGG) were searched against these reads, and medaka_consensu (v1.2.1, option: -m r941_min_high_g360, https://github.com/nanoporetech/medaka) was used for the local assembly of reads with telomeric sequences. Locally assembled contigs were aligned with the original telomere regions using nucmer (v3.1, https://mummer.sourceforge.net/) and were used to replace the original sequences.

The locations of centromeric regions were estimated according to the frequency of all candidate centromeric tandem repeats[21]. Using bedtools, TRF (short tandem repeats) coverage and gene coverage were calculated with a window of 100 K, and the results are displayed as histograms.

### Repeat, noncoding RNA, and gene annotation

We identified de novo repetitive sequences in the genomes of the three species using RepeatModeler (v.1.0.4, option: −pa 10, https://github.com/rmhubley/RepeatModeler). After combining known repetitive sequences of the RepeatMasker library and the de novo repetitive sequences constructed using RepeatModeler, we used RepeatMasker (v.4.0.5, https://github.com/rmhubley/RepeatMasker, options: -nolow -no_is -norna -parallel 4) for repeat annotation. Noncoding RNAs of the three *Stephania* species were annotated using multiple databases and software packages. The tRNA genes were identified using tRNAscan-SE (v1.23, https://github.com/UCSC-LoweLab/tRNAscan-SE). The rRNA fragments were predicted using the rRNA database. Noncoding RNA sequences were predicted using INFERNAL (v1.1.2, https://github.com/EddyRivasLab/infernal, options: −Z 747.66 --cut_ga --rfam --nohmmonly --cpu 15) against the Rfam database.

Gene models for the assembly of the three *Stephania* genomes were predicted according to transcript mapping, ab initio gene prediction, and homologous gene alignment. ONT cDNA reads were aligned against the genome using Minimap2 (v2.17, https://github.com/lh3/minimap2). Transcripts were assembled using stringtie2 (v2.1.5, https://github.com/skovaka/stringtie2, option: -p 15), and the open reading frames of all assembled transcripts were predicted using TransDecoder (v5.1.0, https://github.com/TransDecoder/TransDecoder). Augustus (v3.3.2, https://github.com/Gaius-Augustus/Augustus, options: --uniqueGeneId=true --noInFrameStop=true --gff3 = on --strand=both), Genscan (v1.0, http://bioinf.uni-greifswald.de/webaugustus/predictiontutorial), and GlimmerHMM (v3.0.4, options: -f -g, http://ccb.jhu.edu/software/glimmer/index.shtml) were used for ab initio gene prediction. For homologous gene alignment, proteins from four species (*Arabidopsis thaliana*, *C. chinensis*, *Papaver rhoeas*, and *Aquilegia viridiflora*) were aligned to the genome using Exonerate (v2.4.0, options: --model protein2genome --showtargetgff 1, https://github.com/nathanweeks/exonerate). Finally, we used MAKER (v2.31.10, http://yandell.topaz.genetics.utah.edu/cgi-bin/maker_license.cgi) to integrate the gene sets predicted using the three methods and remove incomplete genes and genes with too-short CDS lengths ( < 150 bp); thus, a nonredundant gene set was obtained. We used BUSCO (v.5.2.2, https://busco.ezlab.org/) to evaluate the prediction quality based on the Embryophyta database.

Functional annotation of the predicted protein-coding genes was performed by applying BLASTP (option: -evalue 1e$^{-05}$) searches against entries in both the National Center for Biotechnology Information (NCBI) NR and UniProt databases (http://www.uniprot.org/). Searches for gene motifs and domains were performed using InterProScan (v5.33, https://github.com/ebi-pf-team/interproscan) and HMMER (v3.1). Gene Ontology terms (http://geneontology.org/) for genes were obtained from the corresponding InterPro (https://github.com/ebi-pf-team/interproscan) or UniProt entry (https://www.uniprot.org/). Pathway annotation was performed using KOBAS (v3.0, https://github.com/xmao/kobas) against the KEGG database.

### Phylogenetic analysis and divergence time estimation

All protein sequences of the fourteen selected species were aligned using diamond (v2.0.4.142, options: --more-sensitive -p 1 --quiet -e 0.001 --compress 1), and gene family clustering was performed using OrthoFinder (v.2.3.12, https://github.com/davidemms/OrthoFinder, option: -M msa). Single-copy gene families shared by all fourteen species were screened to construct phylogenetic trees. A phylogenetic tree was constructed using RAxML (v.8.2.10, https://cme.h-its.org/exelixis/web/software/raxml, options: -f a -p 12345 -x 12345 -# 100), with 100 replicates of bootstrapping. To investigate the divergence times of these plants, we used MCMCtree (options: burnin = 300,0000, nsample = 8,000,000) with calibrations for estimation[60]. Published divergence times for *Oryza sativa* and *V. vinifera* (125-150 Mya) were used for calibration.

To investigate gene family expansion and contraction in *Stephania*, we compared the three *Stephania* genomes with the genomes of 11 other plant species[61]. Contraction and expansion analyses were

performed using CAFÉ (v.3.1, https://github.com/hahnlab/CAFE, options: -p 0.05 -t 8 -r 10,000 -filter) based on the clustering results.

## Whole-genome duplication analysis

Most angiosperms have undergone WGD. WGDs are usually identified from Ks-based age distributions of paralogs (where Ks is the number of substitutions per synonymous site) or from gene collinearity data. All amino acid sequences were self-aligned using BLASTP (v.2.6.0, options: -evalue 1e-05 -outfmt 6), and the best BLASTP results were retained. To obtain paralogous gene families, we performed gene cluster analyses based on CDS alignment using OrthoFinder (v.2.3.12; option: -M msa) and determined syntenic blocks using MCScanX (https://github.com/wyp1125/MCScanx, options: -a -e 1e-5 -s 5 -m 25). Ks values were calculated from all paralogous families using yn00 in the PAML package[60].

## In silico identification of topologically associating domains

TADs were identified using HiC data[62]. Briefly, Hi-C read pairs were mapped to *S. japonica* and *S. cepharantha* genomes, and the contact matrixes were created using HiC-Pro (v 3.1.0, https://github.com/nservant/HiC-Pro). Hi-C contact matrixes with different resolutions were imported into HiCExplorer[63] (v3.7.2) and transformed by the built-in tool hicConvertFormat. After normalization using hicNormalize (options: --correctionMethod KR) and correction using hicCorrectMatrix (options: --filterThreshold -1.5 5), hicFindTADs was used to find TAD with different resolutions (options: --minDepth 5 --maxDepth 10 --step 2 --thresholdComparisons 0.01). The BGCs were predicted using plantiSMASH[23] (v1.0, options: -taxon plants -c 47 --min-domain-number 1).

## Homologous search of enzymes involved in cepharanthine biosynthesis

To investigate the genes involved in cepharanthine (1) biosynthesis, a series of orthologous gene protein sequences were retrieved from NCBI, including NCS, OMT, and cytochrome P450 (CYP450). Using these orthologs as queries, enzyme genes in *S. japonica*, *S. yunnanensis*, and *S. cepharantha* were identified using BLASTP (option: e-value 1e−10). Subsequently, the physical locations of homologous genes clustered in a clade were analyzed and visualized using TBtools (v1.098761, https://github.com/CJ-Chen/TBtools). Additionally, the phylogenetic tree of NCS, OMT, and the CYP450 gene family was constructed with 1,000 bootstrapping replicates. Combined with transcriptome data and phylogenetic tree analysis, we screened 11 NCSs (ScepNCS1-2, SjapNCS1-4, and SyunNCS1-5) for the enzymatic activity assay.

## Metabolomics and metabolite quantification

To investigate the content of BIAs in different tissues of the three *Stephania* species, a ultra-performance liquid chromatography–tandem mass spectrometry system (SCIEX TripleTOF 6600 + ), ultra-high–performance liquid chromatography (ultra-HPLC) coupled with time-of-flight mass spectrometry was used for the relative quantification of metabolites. Briefly, root, stem, and leaf samples of the three species were harvested and freeze-dried in a vacuum freeze drier. All sample powders were weighed (30 mg) and sonicated for 1 h with 1.5 mL of 75% aqueous methanol containing 5 µg/mL umbelliferone as the internal standard. Subsequently, all samples were centrifuged at 13,400 × g for 10 min, and the supernatants were collected and filtered through a 0.22-µm membrane filter. The injection volume of each sample was 1 mL, and the detection wavelength was 282 nm. The column used for separation was a Kinetex C18 100 A analytical column (4.6 mm × 150 mm, 2.6 µm) maintained at 30 °C. Mobile phases A (H$_2$O + 0.1% formic acid) and B (acetonitrile) were run in the following gradient program at 0.4 mL/min: 0–1 min, 10% B; 1–11 min, 10%–95% B; 11–12.5 min, 95% B; 12.5–12.51 min, 95%–10% B; and 12.51–13 min, 10% B. The mass spectrometer was used in the full scan mode at a scan time of

35 ms per transition. Other parameters were as follows: the mass spectrometer was run in the electrospray ionization (ESI) mode in the positive ion mode; spray voltage, 3.5 KV; spray temperature, 550 °C; curtain gas, 35 psi; GAS1, 40 psi; GAS2, 60 psi. The identified metabolites were searched against the Food and Drug Administration–approved drug library from SelleckChem (Catalog #L1300).

## Candidate gene cloning and protein expression

Total RNA was extracted from the three plant species with TRIzol™ Reagent (Ambion, USA). cDNA was synthesized from total RNA using TransScript® One-Step gDNA Removal and cDNA Synthesis SuperMix (TransGen Biotech, Beijing, China). Genes of interest were amplified using PCR from cDNA with gene-specific primers and inserted into the pET28a vector using ClonExpress® II One Step Cloning Kit (Vazyme, China). Unsuccessfully cloned genes were also directly synthesized and inserted into the pET28a vector (Tsingke Biotechnology, China). Finally, recombinant plasmids of NCS were individually introduced into *Escherichia coli* BL21 (DE3). Gene-specific primers are listed in Supplementary Data 1.

*E. coli* cells containing each recombinant plasmid were grown at 37 °C in the Luria Bertani medium containing 50 µg/mL kanamycin and were subjected to induction with 0.3 mM isopropylthiogalactoside at 16 °C. Cells containing the NCS recombinant plasmid were incubated under shaking at 24 × g for 24 h. The cells were collected by centrifugation (2376 × g for 10 min at 4 °C) and resuspended in buffer (50 mM Tris-HCl, pH 7.4; 5 mM β-mercaptoethanol; 10% glycerine). After sonication, the supernatant was collected as a crude enzyme by centrifugation at 13,400 × g for 15 min at 4 °C.

## Enzyme activity assay of NCS and HPLC analysis

The enzyme activities of NCS were determined in buffer (50 mM Tris-HCl, pH 7.4; 5 mM β-mercaptoethanol; 10% glycerine) containing the substrates 4, 5, and 80-mL crude enzyme. After incubation at 37 °C overnight, reactions were terminated by adding 1/2 volume of methanol. The reaction products were centrifuged at 13,400 × g for 15 min, and the supernatant was detected using HPLC analysis.

The column applied for analysis was a Hypersil GOLD™ C18 25005-254630 (4.6 × 250 mm, 5 mm) on a Shimadzu LC-2050C 3D system, with the temperature having set at 35 °C. A 10-mL sample was injected for HPLC analysis, and the detection wavelength was 282 nm. For NCS assay, mobile phases A (H$_2$O + 0.1% formic acid) and B (acetonitrile) were run in the following gradient programs at 0.5 mL/min: 0–10 min, 5% B; 10–15 min, 5%–20% B; 15–15.1 min, 20%–50% B; 15.1–25 min, 95% B; and 25–30 min, 5% B.

## Molecular docking

AutoDockVina[64] was used to analyze the binding affinities and modes of interaction between eight BIA candidates and two targets. The molecular structures of BIAs were retrieved from the NCBI PubChem library (https://pubchem.ncbi.nlm.nih.gov/). The 3D coordinates of 7VX4 (ACE2-RBD in SARS-CoV-2 Beta variant S-ACE2 complex) and 7VX5 (ACE2-RBD in SARS-CoV-2 Kappa variant S-ACE2 complex) were downloaded from the Protein Data Bank (http://www.rcsb.org/pdb/home/). AutoDockTools (https://autodock.scripps.edu/) was used to remove water molecules and other ligands from the protein files and to convert the protein structure files to the PDBQT format. All compound files were converted to PDBQT format using Open Babel[65] (v3.1.1). The grid box was set to 76 Å × 98 Å × 116 Å (7VX4) and 70 Å × 94 Å × 118 Å (7VX5) to cover the structural domain of the protein.

## Antiviral experiments

To evaluate the antiviral activity of BIAs, we determined the inhibitory activity of compounds in the cepharanthine (1) biosynthetic pathway

or other cepharanthine (1) analogs against SARS-CoV-2 live virus, SARS-CoV-2 trVLP, SARS-CoV-2 alternative models GX_P2V, SADS-CoV, and PEDV in different cell lines.

## Cell lines, coronavirus, and key reagents

All these BIAs in the antiviral assays were purchased from different companies (Supplementary Table 7), and dissolved in DMSO (Solarbio, Beijing, China) at 10 mM. 30 (TargetMol, catalog No. T7766) was dissolved in DMSO (Solarbio, Beijing, China) at 10 mM.

Caco 2-N cells (kindly provided by Prof. Ding Qiang of Tsinghua University), Vero E6 cells (NICR, Beijing, China) and Huh7 cells (NICR, Beijing, China) were grown in Dulbecco's modified Eagle medium (Gibco, Carlsbad, CA, USA) containing 10% fetal bovine serum (PAN, Aidenbach, Germany) and 1% antibiotic/antimycotic (Gibco, Carlsbad, CA, USA). BHK21-ACE2 cells is kindly provided by Huan Xu in Shenzhen Bay Laboratories.

SARS-CoV-2 trVLP was kindly provided by Prof. Ding Qiang of Tsinghua University and propagated in Caco 2-N cells. In the SARS-CoV-2 trVLP/Caco-2-N system, only the SARS-CoV-2 N gene was replaced by the GFP reporter gene while the engineered Caco-2-N cell line stably expressed and supplemented the viral N protein for live virion package. Previous studies have verified that the SARS-CoV-2 trVLP highly mimicked the transcriptional and replication processes of SARS-CoV-2 and was a live and infectious virus model to test the anti-SARS-CoV-2 efficacy of drug candidates in a BSL-2 laboratory[30]. SARS-CoV-2 trVLP was not a pseudovirus but a live virus model that was used to test the anti-SARS-CoV-2 abilities of drugs[30]. This model has been widely used in published studies[20,66–69]. SARS-CoV-2 trVLP is scientifically reliable model to evaluate the activity of drugs against SARS-CoV-2. GX_P2V (accession no. MT072864.1) was isolated in Vero E6 cells from a dead smuggled pangolin in 2017[28] and propagated in Vero E6 cells. PEDV strain CV777 (accession no. AF353511.1) and SADS-CoV strain CN/GDWT/2017 (accession no. MG557844) were maintained and amplified in Huh 7 cells. SARS-CoV-2 (SARS-CoV-2/WH-09/human/2020/CHN; GenBank: MT093631.2) was maintained and amplified in Vero E6 cells in the biosafety level 3 lab.

## Viral infection assay in vitro and determination of EC50

We evaluated the anti-SARS-CoV-2 trVLP activity using Caco-2-N cells, anti-GX_P2V activity using Vero E6 cells, anti-PEDV activity and anti-SADS-CoV activity using Huh 7 cells. The antiviral assays of compounds were performed according to previous reported method[5] (Supplementary Table 8). Briefly, cells ($2\times10^4$ cells per well) were seeded into 96-well plates and cultured overnight. Subsequently, the cells were treated with a mixture solution of twofold serially diluted drugs and virus at an MOI of 0.01. At 2 h post-infection, the supernatant was removed, and fresh culture medium containing the same concentration of drugs was added to each well. Viral RNA in cell lysates was collected at 48 hpi and quantified using quantitative reverse transcription–PCR (qRT-PCR). The concentrations of drugs against different coronavirus are presented in Supplementary Data 1.

Relative expression levels of viral RNA were normalized according to GAPDH and calculated using the $2^{-\Delta\Delta Ct}$ method[70]. The inhibition rate was calculated using the following equation:

$$\text{Inhibition}(\%) = \left(1 - \frac{\text{relative expression of viral RNA in drug treatment}}{\text{expression of viral RNA in virus control}}\right) * 100\% \tag{1}$$

The drug concentrations for 50% of the maximal effect (EC50) was analyzed using GraphPad Prism 8 (GraphPad Software Inc., San Diego, CA, USA).

## EC50 assay of antiviral activity against SARS-CoV-2 live virus

Cells ($5 \times 10^3$ cells per well) were seeded into 96-well plates and cultured overnight. Subsequently, the cells were treated with a mixture solution of twofold serially diluted drugs and virus at an MOI of 0.01. At 2 h post-infection, the supernatant was removed, and fresh culture medium containing the same concentration of drugs was added to each well. Cytopathic effect (CPE) was observed under light microscope at 48 hpi Cells with CPE changes were marked as "+", cells without CPE changes or normal cell morphology were marked as "-". The EC50 was analyzed using the following equation:

$$\text{EC50} = \text{Antilog}\left(D + \frac{50 - B}{A - B} * C\right) \tag{2}$$

A: Percentage greater than 50% inhibition; B: percentage less than 50% inhibition;

C: log (dilution); D: log (drug concentration less than 50% inhibition).

## Time of addition

The time of addition assay was performed[71]. Vero E6/Caco 2-N cells were seeded into 48-well plate ($2 \times 10^5$ cells/well) and incubated overnight. For "Full-time" treatment, Vero E6/Caco 2-N cells were treated with the drugs and GX_P2V (MOI = 0.01)/SARS-CoV-2 trVLP (MOI = 0.01) for 2 h at 37 °C. Afterwards, the virus–drug mixture was removed, and the cells were cultured with drug-containing medium until the end of the experiment. For "Entry" treatment, the drugs and GX_P2V (MOI = 0.01)/SARS-CoV-2 trVLP (MOI = 0.01) were added to the Vero E6/Caco 2-N cells for 2 h at 37 °C, and then the virus–drug mixture was replaced with fresh culture medium and maintained till the end of the experiment. For "Post-entry" treatment, drugs were added at 2 hpi, and maintained until the end of the experiment. Virus yield in the infected cell lysates was quantified by qRT-PCR and NP expression in infected cells was analyzed by Western blot at 12 hpi.

## Viral RNA extraction and real-time polymerase chain reaction (qRT-PCR)

RNA extraction was performed using the Tissue/Cell Total RNA Extraction Kit (NOBELAB, Beijing, China), according to the manufacturer's instructions. Reverse transcription was performed with the HiScript II Q RT SuperMix for qPCR ( + gDNA wiper; Cat. no. R223-01; Vazyme, Nanjing, China), and qRT-PCR assay was performed using the QuantStudio 1 Real-Time PCR detection system (Applied Biosystems, Foster City, CA, USA) with Taq Pro Universal SYBR qPCR Master Mix (Cat. no. Q712-2; Vazyme, Nanjing, China). The relevant qRT-PCR primer sequences are shown in Supplementary Table 9.

## Drug cytotoxicity assays and determination of CC50

The cytotoxicity of different drugs on Caco 2-N/Vero E6/Huh-7 cells was measured using the CellTiter-Blue® assay. Particularly, Caco 2-N/Vero E6/Huh-7 cells were plated in 96-well plates ($2 \times 10^4$ cells per well). After overnight incubation, different concentrations of drugs were added, followed by incubation for 48 h. Subsequently, 20 μL of Cell-Titer-Blue® Reagent (G8081; Promega, Madison, WI, USA) was added to each well, and the plate was incubated at 37 °C for 1–2 h. Cell viability was assessed by detecting the resorufin fluorescence signal (excitation wavelength: 554 nm, and emission wavelength: 593 nm). The cytotoxicity rate was calculated using the following equation:

$$\text{Cytotoxicity}(\%) = \left(1 - \frac{\text{fluorescence vaules(drug treatment} - \text{blank)}}{\text{fluorescence vaules(control} - \text{blank)}}\right) * 100\% \tag{3}$$

The cytotoxicity concentration of 50% of the compounds (CC50) was analyzed using GraphPad Prism 8 (GraphPad Software Inc., San Diego, CA, USA).

## Pseudotyping of VSV and pseudotype-based inhibition assay

SARS-CoV-2 pseudotyped virus was produced[31]. Briefly, pcDNA3.1.VSVG (Addgene ID: 158528) plasmid, pcDNA3.1.S2 plasmid (Addgene ID: 149457) and the plasmid expressing the SARS-CoV (NC_004718.3)/MERS-CoV (NC_019843.3) S protein (Tsingke Biotech, Shanghai, China) were extracted with endotoxin-free reagents. HEK-293T cells were transfected with the plasmid pcDNA3.1.S2 and the plasmid pcDNA3.1.VSVG, after which the G*ΔG-VSV solution was added to amplify the G*ΔG-VSV pseudotyped virus (EH1020-PM). Then, plasmids of the SARS-CoV-2/SARS-CoV/MERS-CoV S protein were transfected into HEK-293T cells to provide membrane proteins on the cell surface, and G*ΔG-VSV infection provided genomes of VSV. After 24 h and 48 h of infection and transfection, the cell supernatants were collected, mixed and centrifuged at $1000 \times g$ for 10 min. The SARS-CoV-2/SARS-CoV/MERS-CoV S protein pseudovirus was obtained after filtration using a 0.45 μM filter. Then, 1 mL aliquots of the solution mixture were distributed into 2-mL microtubes and stored at −80 °C.

To determine the effect of these compounds on viral entry, pseudotyped virus neutralization assays were performed[31]. The anti-SARS-CoV-2/SARS-CoV pseudovirus activity of BIAs was performed on BHK21-ACE2 cells, and the anti-MERS-CoV pseudovirus activity of BIAs was performed on Huh-7 cells. Cells ($5 \times 10^4$ cells per well) were seeded into 96-well plates. Cells and pseudoviruses were separately treated with different concentrations of drugs for 1 h at 37 °C and 4 °C, respectively, before pseudoviruses infection. After 24 hours of infection, luciferase expression was quantified by measuring the relative light unit (RLU) using a microplate reader (H1; Bio-Tek) and a luciferase detection kit (Cat. no. 11401ES60; Yeasen Biotech, Shanghai, China). The inhibition rate was calculated using the following equation:

$$\text{Inhibition}(\%) = \left(1 - \frac{\text{RLU(drug treatment} - \text{blank)}}{\text{RLU(virus control} - \text{blank)}}\right) * 100\% \quad (4)$$

The drug concentration for 50% of the maximal effect (EC50) was analyzed using GraphPad Prism 8 (GraphPad Software Inc., San Diego, CA, USA).

## Western blotting analysis

Proteins were harvested with radio immunoprecipitation assay (RIPA) lysis buffer containing PMSF (1 mM) (Beyotime, Shanghai, China), and boiled at 100°C for 10 min together with loading buffer (TransGen Biotech, Beijing, China) after concentration determination. Subsequently, the samples were loaded on a 10% SDS-PAGE gel and transferred to a polyvinylidene fluoride (PVDF) membrane (Millipore, Burlington, MA, USA) using Trans-Blot SD semi-dry transfer cell (Bio-rad, Hercules, CA, USA). Antibodies against nucleocapsid protein of SARS-CoV-2 (GenScript, catalog No. A02049-100, Nanjing, China) and GAPDH (Proteintech, catalog No. 60004-1-Ig, Wuhan, China) were used at 1:1000 and 1:10,000 dilutions, respectively. The second antibody of HRP-conjugated AffiniPure goat anti-mouse IgG (H + L) (Proteintech, catalog No. SA00001-1, Wuhan, China) was diluted at 1:10,000. Then Immobilon Western Chemiluminescent HRP Substrate (Millipore, Burlington, MA, USA) and Tanon-5200 Multi Gel Imaging Analysis System (Tanon, Shanghai, China) were used for imaging.

## Reporting summary

Further information on research design is available in the Nature Portfolio Reporting Summary linked to this article.

## Data availability

All raw sequencing data were deposited under the National Center for Biotechnology Information (NCBI) GenBank accession number PRJNA888087. The three *Stephania* genome assemblies and annotations generated by this study have been archived under the China National GeneBank DataBase (CNGBdb) accession CNP0003595. The information of functional BIAs-biosynthetic genes and compounds are available at GitHub [https://github.com/liuzy2008/evo-chemo_anti-SARS-CoV-2_drug_discovery2023]. Source data are provided with this paper.

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

## Acknowledgements

We acknowledge helpful discussions and feedback from Yujun Zhang (Institute of Chinese Materia Medica, China Academy of Chinese Medicine Sciences), Yong E. Zhang (Institute of Zoology, Chinese Academy of Sciences), Kai Chen (Kunming University of Science and Technology), Ke Zhang (Shanghai Institute of Immunity and Infection, Chinese Academy of Sciences), and Kai He (Guangzhou University). This work was sponsored by the National Key Research and Development Program of China (2023YFC3504800 to Z. Xu, 2020YFA0712102, BWS21J025, and 20SWAQK22 to H. Fan,), the National Natural Science Foundation of China (82151224 to Y. Tong, 82202492 to H. Fan, and 82274037 to Z. Xu), the H&H Global Research and Technology Center (H20230550 to H. Fan), and the Fundamental Research Funds for the Central Universities (QNTD2023-01 to H. Fan).

## Author contributions

L. Leng, Z. Xu, H. Fan, C. Song, Y. Tong, and S. Chen designed the project; L. Leng, Z. Xu, B. Hong, B. Zhao, Y. Tian, C. Wang, L. Yang, Z. Zou, L. Li, K. Liu, W. Peng, J. Liu, Z. An, Y. Wang, B. Duan, Z. Hu, M. Li, and Z. Liu performed experiments and analysis; B. Duan, Z. Hu, and X. Li collected the materials; L. Leng, Z. Xu, H. Fan, and C. Song wrote the original draft; L. Leng, C. Wang, Z. Bi, T. He, and B. Liu designed figures; L. Leng, Z. Xu, C. Zheng, S. Zhang, H. Fan, C. Song, Y. Tong, and S. Chen reviewed and edited the manuscript; H. Fan, C. Song, Y. Tong, and S. Chen supervised the project. All authors have read and approved the manuscript for publication.

## Competing interests

The authors declare no competing interests.
