## [Peer Review File · Nature Communications]

Cepharanthine analogs mining and genomes of *Stephania* accelerate anti-coronavirus drug discoveryReviewers' Comments:

Reviewer #1:

Remarks to the Author:

Leng et al., have made great efforts in sequencing and assembling three *Stephania* genomes, which encode expanded cepharanthine related compounds that have been shown to possess anti-viruses activity. They also demonstrated that only cepharanthine analogs but not any biosynthetic intermediates presented anti-coronavirus properties, which may suggest a new strategy by using Nature-generated chemical diversity to fish for drugs in the post-pandemic era.

While I do like the idea of exploring already-evolved chemicals to develop new drugs, I am afraid this does not fall into the area of Darwinian medicine as the authors have stated across this manuscript. Although evolutionary medicine refers to the applications of the principals of evolution to medical fields, in this m/s I feel that the authors have mis-used this term because plants have evolved the chemical diversity, in most of the case, is for its own adaptation other than for the defense of virus that infect human beings. Developing drug analogs based on activity-known compounds is a general strategy that have been used for a long time, I do not see how "creative" it is in this m/s by searching similar compounds in closely related species. Besides, the authors have assembled three high-quality genomes, but they haven't explored the genomic basis (e.g. gene duplication/loss, gene clustering) underpinning the chemical diversity around cepharanthine (which actually perfectly reflects the idea of "evo-chemo").

Reviewer #2:

Remarks to the Author:

The authors focused on the anti-SARS-CoV-2 activity of cepharanthine and aim to identify agents with higher potency and broad spectrum antiviral activity. They assembled three *Stephania* genomes and identified the putative enzymes involved in cepharanthine biosynthesis. They also tested the antiviral activity of 23 cepharanthine-related compounds using GX_P2V and SARS-CoV-2 model viruses. The activity of these compounds was diverse depending on the assay systems. The authors also found five cepharanthine analogs derived from *Nelumbo* and *Thalictrum*, which may be potential sources for identifying highly potent antiviral compounds.

The subject of the study is of academic and clinical importance and the concept of evo-chemo strategy for identifying antiviral compounds is interesting. However, the description of the experiments in this paper was poor and is not understandable. For example, how all the compounds were prepared? What's the experimental condition of the infection assay, including the strain name of the viruses and the assay system used in Fig. 5A? Although the Y-axis in all the graphs in Fig. 3 indicates the percentage of viral levels, it seems that those should show the inhibition percentage since the values increase along with the concentration of compounds. In addition, evaluation of antiviral activity of the compounds in infection assay was inconsistent between the assay systems as shown below and was technically doubtful.

(major points)

- In Fig. 3, since the antiviral activity of compounds was different depending on the virus used in the assay (especially, berbamine, dauricine, and daurisoline), the authors should evaluate the activity in SARS-CoV-2 infection assay, rather than VLP and pseudotyped SARS-CoV-2. Without this assay, anti-SARS-CoV-2 activity cannot be evaluated.
- In Fig. 3E, dauricine at 6.25 μ M with full-time showed almost 100% inhibition, although its EC50 was calculated as high as 15.73 μ M in Fig. 3A, which is really strange. Same thing happened on daurisoline.
- Isotetrandrine at 50 μ M and daurisoline at 12.5 μ M had not big effect on the entry in Fig. 3F. If it is the case, these compounds should not affect on SARS-CoV-2 pseudo typed virus in Fig. 3C, which evaluates SARS-CoV-2 entry. But isotetrandrine and daurisoline showed high anti-entry activity with

EC50 of 7.32 and 8.48 μM , respectively. It is logically unreasonable.

- As well, the cytotoxicity of tetrandrine was inconsistent between Fig. 3A and 3B.

- Although the authors aim to identify broad spectrum antiviral compounds, they only test on two viruses, GX_P2V and SARS-CoV-2. Only by these data, they cannot claim even for pan-betacoronavirus or anti-coronavirus activity without SARS-CoV-1 and MERS-CoV. Demonstrating broad spectrum antiviral activity requires infection assay using plus-stranded, minus-stranded, retroviruses, etc. other than coronaviruses.

- The authors' second aim to identify agents having significantly higher antiviral activity than cepharanthine was not achieved either, based on the results shown in Fig. 3C and 3B.

Reviewer #3:

Remarks to the Author:

The manuscript by Leng et al. described the genome assembly of three *Stephania* species, and evaluated a number of monomeric and dimeric BIA alkaloids produced by these plants for their activity against SARS-Cov2 virus. I find these discoveries provide valuable data for the natural product field to further embark on. Some BIAs showed micro molar range of effective dose EC50 values, indicating that they their structure can be further developed into useful antiviral drugs. The manuscript itself is very well written.

There are several areas that may need substantial improvement for this study:

1) The authors suggested a "evo-chemo" strategy to accelerate drug discovery. At present, there is no support for this claim from this study. Structurally similar compounds are expected to have similar biological functions, known as structure activity relationship. It is possible to evaluate the bioactivities of all the intermediates through the discovery of a biosynthetic pathway. However claiming this as a "evo-chemo" strategy is ambitious, and will need substantial experimental supports from either this study or other studies. So far the authors only have this as a proposal, which did not seem to have accelerated the antiviral drug discovery or be relevant in this manuscript.

2) While the BIAs showed anti SARS-Cov2 activity, it is unclear the mode of mechanism. It may be difficult or require a new study to investigate the inhibition mechanism, but at the current stage, the antiviral study lacks a solid conclusion. Not having a clear mechanism also did not help convey the key message of an "evo-chemo" strategy to accelerate drug discovery.

3) The authors used the genome assemblies to deduce a number of putative genes for BIA biosynthesis. However none of the putative genes were tested, and they remain putative. If the authors could show the biosynthesis of some BIA intermediate with their putative genes, and combine the anti-viral studies using these intermediates, it may help to substantiate some claims in the "evo-chemo" strategy the authors proposed.

Reviewer #4:

Remarks to the Author:

In the presented article, titled, "The evo-chemo strategy accelerates anti-SARS-CoV-2 drug discovery", Leng et al. described a novel strategy to discover new metabolites that could be the potential medicine with broad-spectrum anti-viral properties. Authors reported three plant genomes, one (*Stephania japonica*) being a high quality genome with a single gap, performed some metabolome analysis using these plant species, and then coined the term "evo-chemo" strategy to say the results that they have arrived using comparative genomics and analyzing metabolites for anti-viral properties. I feel that the genome assemblies created in this study is good, but that is the only credit I will give to this study. Authors seems absolutely disillusioned, and the study design is everywhere. The results keep jumping

everywhere, which is not coherent. Further, I have no idea why authors think that this so called "evo-chemo" is a new strategy, but rather it is just a term that they wish to coin. I do not see any novelty what so ever that may help to discover new drug candidates. The manuscript title is misleading, and since the objective of this study is not clear, authors seems have no clue what they want to report, which makes this study with otherwise nice genome assemblies not impactful. In my opinion, this study is not worth publication at Nature communication in its present form. The datasets are impressive, but the treatment of the datasets are directionless and misleading at occasions in my opinion. Each result section seems incomplete with many questions remaining unexplored. I am listing below reasons, specific comments and my opinion on this study, with the hope that authors find them constructive and could use it for the improvement of this study-

1. I disagree with the title and the abstract (the last two sentences). The so called evo-chemo strategy is nothing and what people have been using for many many years.
2. Please provide real picture of the plants (Fig 1D). Any specific reason to show roots in the *S. cepharantha*?
3. Please provide chromosome names that has been used for specific plant genomes to the contact maps (Fig 1B). In figure 2B, authors show chromosome 1 of *S. japonica* having relationships with two chromosomes of *Syun* and *Scap*. What is chromosome 1 of *Sjap* in the contact map? In the contact map, fifth block from the bottom for *S. japonica* shows some strange structure, please comment on that. Also, please The box boundary is missing for third box for *S. japonica* and 7th box for *S. yunnanensis* in Fig 1B. Also, Please provide karyotyping data here. I think that this data will ensure that the chromosome numbers matches with pseudo molecules reported in this study. Even a reference of chromosome numbers will be good enough. Further, *S. japonica* has 11 chromosomes, two of which (chromosome 1 and chromosome 2) seems to have associated with four chromosomes of *Syun* and *Scep*, resulting in 13 chromosomes. So, what happens first? I mean, the two chromosomes of *Sjap* resulted in four chromosomes of *Syun* and *Scep*, or the fusion of two chromosomes resulted in a single chromosome of *S. japonica*. Please consider talking about this interesting aspect of the genomes.
4. Please provide more information about the experimental design in the introduction. I mean, why you choose *Sjap*, *Syun* and *Scep*? Three plants from the same genus and same phylum to do what? Why not choose plants from same genus that are not producing the metabolites with medicinal properties. In that case, the resource will help in answering a lot of interesting questions. In this study, authors would have concluded exactly the same thing using *S. japonica* genome assembly.
5. Please provide more information about the cepharanthine and intermediates, what is known, what is not known and so on. Any FDA approved drug for this molecule or its derivative? What plant families produces this compound. Does intermediates are previously reported? In the abstract section, authors mentioned about seven of nine cepharantine analogs with anti-coronavirus properties, so these metabolites were not know before? These are the new metabolites reported for the first time ever? This aspect is completely unclear. Authors need to provide more details as where these analogs have previously reported and so on. Authors mentioned about *Nekumbo* and *Thalictrum*, and that they produces ceptharanthine analogs, so, these are different from *Stephania*? How these are different? Also, why authors opted to sequence three genomes from *Stephania* but not from *Nekumbo* and *Thalictrum*. Experimental design wise, include plants from these families would have provide much important insights on the biosynthesis pathways. This is a major weakness.
6. In the introduction, authors mention objective in page 6, from line 148 till line 151. I disagree as genome sequencing and comparative genome analysis has been the way forward to answer exactly this question for several years. In these previous studies, intermediates have been identified across multiple plant species, and comparative genome analysis have explained the emergence of specialized metabolites. Genome studies of *Opium poppy*, *Campotheca acuminata*, *Ophiorrhiza pumila*, and so many other medicinal plant species are example. So, the objective is not really an objective, rather misleading.
7. Line 181, "...species and constructed.....", what is strictly single-copy gene. Single copy gene is single copy gene. This is not clear. Please clarify.
8. Line number 184, authors says that *S. japonica* belongs to the subgenus *Stephania*, and the rest of the two plants belong to subgenus *Tuberiphania*, then why authors keep using *Stephania* for all the

three plant genomes. It does not make sense and very confusing. I am not sure what classification authors are using. Authors have used *Stephania* to describe the plant species together, but then if they belong to different subgenus, then why to use this way?

9. Line 203 to line 207, authors mention that *S. japonica* has not undergone shared whole genome triplication that is shared between core eudicots, and therefore, they reported 2:3 synteny ration between *S. japonica* and *V. vinifera*. This is reasonable. But then how authors can say that *Stephania* species diverged from *V. vinifera*? The synonymous substitution plot (Fig 1E) is very hard to understand. Please consider choosing the color properly, and if possible or like, separate the orthologs and paralogs sets.

10. Line 198, it says that *S. japonica* genome assembly is gap free, but it has a single gap. Please correct this point.

11. Unless I have missed, please provide a supplementary table with genes that were used to identify homologs for NCS, methyltransferases, and *cyp450*. Also, if possible, create a github repository for this manuscript, and provide the sequences used here. Further, in line 252, authors mention about 267, 331 and 234 potential enzymes, so, these numbers of NCS, methyltransferases and *cyp450* combined for a specific plant species.

12. Authors describe BCGs from Line 260 onwards. Here, authors talk about the expression of genes that are conserved and part of BCGs. But I am not sure why authors describe *Sjap* and *Scep*, and not *Syun*? In principle, the three genomes are highly conserved, in which, *Scep* and *Syun* is closer and have higher synteny compared with *Sjap*. So, how many BCGs are conserved across all these three species? Is expression is consistent between *Scep* and *Syun*? What tissue is well known to produce the key medicinally relevant metabolite? Does the expression is consistent in BCGs for tissues which produces high level of key metabolites? Does the tissue that accumulated the key medicinal BIAs are same for all the plant genomes? These are basic questions that is completely ignored which other wise would provide interesting aspects of BIA dimer biosynthesis. The comparative genome analysis between these three plants are ignored, and at times, not consistent. Like, in this case, the comparison is described between *Sjap* and *Scep*, but *Syun* is completely ignored, and as a reader, I wonder why? Authors talk about co-inheritance between two species, but what happens when *Syun* is included? "incompletely consistent" is not the proper way to mention.

13. I recommend authors to consider drawing a graph based pangenome for the *Stephania* species. I understand that this is only three genome, but it will be nicer to see as what constitutes core genome, what is unique to each species, and if some functional aspect comes up through this. Since the genome is nice, authors could even identify the structural variants within individual genomes, and then see the similarity and differences that exists in terms of the distribution of these variants across the three plant genomes described here.

14. Authors mentioned that several BIAs from *Stephanias* had been previously reported. Any new metabolites that authors detected in this study? If not, then these metabolites were already known, right? How authors confirmed the metabolites identity, using standards or MS/MS analysis? How authors excluded the false positives? The proposed pathway for BIAs in *Stephania* species were drawn based on what? The identified metabolites chemical structure were considered to place them in the sequence or the order of the reaction is already know? Again, authors keep talking about cepharanthine biosynthesis pathways, but at occasions, it feels like the pathway is known and at occasions, it feels like this study is proposing the pathway. It is very confusing and authors need to clarify this aspect.

15. For the anti-viral experiment, what is the positive control, daurisoline? Tetrandrine is control for cytotoxicity? Anti-viral activity of cycleanine is never reported before and this is the first time that it was detected? Authors says that the BIA dimers showed paninhibitory activity of BIA dimers, which was also the previous observation. Then what is the novelty of the entire section (Line 331 till 367). Authors could have started this section with line number 368. I am not sure what authors then identified here in this section. Dauricine is used molecule as anti-viral drug?

16. The entire section starting from line 388 is based on assumptions and not fact. Authors could have tested if the BIA dimers have different affinity with the target proteins compared with the monomers through docking. Further, why authors assumed that the dimerization is the decisive step towards anti-viral properties? What is the basis? Previous studies claimed it? If yes, the describe it, and the

refute the claim through your data. Authors does talk about putative genes but they do not test it to confirm, and then this analysis or even discussion becomes meaningless.

17. Section starting from Line 431 seems to me clueless. Why authors choose these plants? Why not perform genome sequencing for selected plants from these different genus and not all three from the same? Authors say that they found five antiviral analogs, from where, gymnosperms, monocots or Amborellales? The placing of this sentence is misleading and I am not able to understand purpose here. Authors says that no BIAs have been reported in gymnosperm, then please use some genomes from gymnosperm to explore biosynthesis pathways for BIAs dimers that authors are describing here. Also, how authors can make this claim? Any reference?

18. I completely disagree with entire section starting from line no 466. This is not a new strategy, and has been in practice since always. Did authors elucidated the pathway here? Authors in principle says that you need to know anti-viral compound, its properties and which plant produces it. Then, analyze the plant, find the intermediates, then then start looking for new metabolites that could have anti-viral activities. But the choice of metabolite is because that was already tested and clinically proven. Then what is the point. You can predict the improved ligand binding through molecular docking, and can guess even as what changes in the chemical can result in better action.

19. I am now sure what is the basis for the authors to claim that " Our result showed that the chemo....(Line 509)". I do not see any result leading to this statement in this article. Similarly, no results is presented to claim line 544, "Our results laid the foundation...". Authors reported genomes, and metabolites were previously reported. What else?

20. Line 707, "a UPLC-MS/MS system..." seems wrong. Please confirm. Is this a triple Q ion trap or Triple TOF? These are two different things. Are they in-line? Why? What is the advantage and how it helped them? What fragments (daughter ion) for a specific metabolites were used for MRM analysis?

Responses to

Reviewer #1 (Comments to the Author):

Leng et al., have made great efforts in sequencing and assembling three *Stephania* genomes, which encode expanded cepharanthine related compounds that have been shown to possess anti-viruses activity. They also demonstrated that only cepharanthine analogs but not any biosynthetic intermediates presented anti-coronavirus properties, which may suggest a new strategy by using Nature-generated chemical diversity to fish for drugs in the post-pandemic era.

Response: Thanks for your positive comments on our manuscript. We have followed your suggestions to improve the manuscript quality.

Q1. The concept of “evo-chemo”

While I do like the idea of exploring already-evolved chemicals to develop new drugs, I am afraid this does not fall into the area of Darwinian medicine as the authors have stated across this manuscript. Although evolutionary medicine refers to the applications of the principals of evolution to medical fields, in this m/s I feel that the authors have mis-used this term because plants have evolved the chemical diversity, in most of the case, is for its own adaptation other than for the defense of virus that infect human beings. Developing drug analogs based on activity-known compounds is a general strategy that have been used for a long time, I do not see how “creative” it is in this m/s by searching similar compounds in closely related species.

Response: We appreciate your constructive suggestions. Accordingly, we have deleted the confusing expressions of Darwinian medicine, and altered the tone of our conclusion. Related descriptions in the revised manuscript are as follows:

“Main:

“Cepharanthine belongs to benzyloisoquinoline alkaloids (BIAs), a group of nearly 2,500 specialized plant metabolites with remarkable pharmacological effects. Several BIAs are approved pharmacological compounds. For instance, cepharanthine, approved by the Pharmaceuticals and Medical Devices Agency (PMDA) in Japan, is used to treat cancer and inflammation. The well-known antibacterial and hypolipidemic compound berberine, is approved by PMDA and the China

Food and Drug Administration (CFDA) to treat infection and parasitology; and tetrandrine has been approved by CFDA as a calcium channel blocker. Given the potent anti-SARS-CoV-2 property of cepharanthine, the investigation of other antiviral cepharanthine analogs becomes compelling, which inspired us to explore such compounds.” (Line 147, page 6)

Discussion:

“The concept of structure-activity relationship, where structurally similar compounds are expected to share similar pharmaceutical functions, has been employed to identify effective antibacterial drugs. In this study, we propose a methodology to identify new candidate compounds from a known potent antiviral natural product (Fig. 7). Our findings indicate that, similar to cepharanthine, a group of cepharanthine analogs possess potential broad-spectrum anti-coronavirus activity. Furthermore, it is crucial to underscore that this group of analogs stems from the same plant clade as that of cepharanthine. We posit an “adaptive” hypothesis to explain this phenomenon.” (Line 588, page 27)

Q2. Genomic basis underpinning the chemical diversity

Besides, the authors have assembled three high-quality genomes, but they haven't explored the genomic basis (e.g. gene duplication/loss, gene clustering) underpinning the chemical diversity around cepharathine (which actually perfectly reflects the idea of “evo-chemo”).

Response: We performed enzymatic experiments with norcoclaurine synthase (NCS). These experiments from two syntenic gene array regions were conducted and the genomic basis behind NCS enzymatic activity was discussed. We have redrawn figure 2 and rewritten the corresponding text as follows:

“We subsequently validated the putative genes for BIA biosynthesis, and all the NCS members from all three species were evaluated, particularly those within two syntenic regions (Fig. 2B, H). These regions encompassed multiple several NCS homologs, forming complete or split gene pairs and arrays. To avoid overlooking possible NCS genes, we conducted local searches for neighboring genes (blast search with E-value = 10^{-10}). The first region comprised seven, seven, and six NCS copies in *S. japonica*, *S. yunnanensis*, and *S. cepharantha*, respectively. The second region contained three, three, and two NCS copies. Catalytic experiments involving 12 NCSs (11 *Stephania* NCSs with high expression and one known-function NCS from *Coptis chinensis* as a positive control) were conducted using dopamine and 4-HPAA as substrates (Fig. 2I). The results demonstrated that five

candidate *Stephania* NCS genes possessed NCS functionality. Remarkably, four out of these five *Stephania* NCSs exhibited superior enzymatic activities compared with the *Coptis chinensis* NCS. Notably, these four NCSs were all situated in the second syntenic region (Fig. 2H, I). Phylogenetic analysis unveiled that these four *Stephania* NCSs formed a monophyletic group, with two *S. yunnanensis* NCSs emerging as paralogs due to a recent duplication event (Fig. 2J). These two *S. yunnanensis* NCSs were highly expressed in stems, whereas ScNCS1 and SjNCS4 exhibited high expression in roots. These results underscore the dynamic evolution of *Stephania* NCS genes in terms of both local copy number adjustments and changes in expression patterns.” (Line 314, page 14)

Fig. 2H-J

Reviewer #2 (Comments to the Author):

The authors focused on the anti-SARS-CoV-2 activity of cepharanthine and aim to identify agents with higher potency and broad spectrum antiviral activity. They assembled three *Stephania* genomes and identified the putative enzymes involved in cepharanthine biosynthesis. They also tested the antiviral activity of 23 cepharanthine-related compounds using GX_P2V and SARS-CoV-2 model viruses. The activity of these compounds was diverse depending on the assay systems. The authors also found five cepharanthine analogs derived from *Nelumbo* and *Thalictrum*, which may be potential sources for identifying highly potent antiviral compounds.

The subject of the study is of academic and clinical importance and the concept of evo-chemo strategy for identifying antiviral compounds is interesting.

Response: Thanks for your positive comments on our manuscript. We have followed your suggestions to improve the manuscript quality.

Q1. Description of the experiments

However, the description of the experiments in this paper was poor and is not understandable. For example, how all the compounds were prepared? What's the experimental condition of the infection assay, including the strain name of the viruses and the assay system used in Fig. 5A? Although the Y-axis in all the graphs in Fig. 3 indicates the percentage of viral levels, it seems that those should show the inhibition percentage since the values increase along with the concentration of compounds. In addition, evaluation of antiviral activity of the compounds in infection assay was inconsistent between the assay systems as shown below and was technically doubtful.

Response: We have revised the description of the experiments more carefully to avoid misinterpretation. (Line 864-884, page 37). Specifically, all the compounds preparation information is summarized in Supplementary Table 9. The strain name of the viruses and the assay system used in Fig. 6A (Fig.5A in the last version) are GX_P2V and SARS-CoV-2 trVLP, respectively, we have described related information in the legends of Fig.6 (Fig.5 in the last version).

The left and right Y-axis of the graphs in Fig. 3 represent mean percentage of inhibition against virus yield and cytotoxicity of BIAs, respectively. And evaluation of antiviral activity of the compounds in

infection assay between the assay systems are carefully described to avoid misunderstanding, and are clarified below in a point-by-point style.

Q2. Infection assay

(major points)

- In Fig. 3, since the antiviral activity of compounds was different depending on the virus used in the assay (especially, berbamine, dauricine, and daurisolone), the authors should evaluate the activity in SARS-CoV-2 infection assay, rather than VLP and pseudotyped SARS-CoV-2. Without this assay, anti-SARS-CoV-2 activity cannot be evaluated.

Response: We appreciate this valuable suggestion. In this study, we used the pangolin coronavirus GX_P2V model, which is similar to SARS-CoV-2 both in terms of genomic and amino acid level, for initial antiviral screening of isoquinoline alkaloids against coronavirus infection, and then SARS-CoV-2 trVLP was used for anti-SARS-CoV-2 activity evaluation. In the SARS-CoV-2 trVLP/Caco-2-N system, only the SARS-CoV-2 N gene was replaced by the GFP reporter gene while the engineered Caco-2-N cell line stably expressed and supplemented the viral N protein for live virion package. Previous studies verified that the SARS-CoV-2 trVLP highly mimicked the transcriptional and replication processes of SARS-CoV-2 and was a live and infectious virus model to test the anti-SARS-CoV-2 efficacy of drug candidates in a BSL-2 laboratory (PLoS Pathog. 2021, 17, e1009439). SARS-CoV-2 trVLP was not a pseudovirus but a live virus model that was used to test the anti-SARS-CoV-2 abilities of drugs. (PLoS Pathog. 2021;17(3):e1009439.). This model has been widely used in different studies (Cell. 2021;184(7):1865-1883.e20., Signal Transduct Target Ther. 2023;8(1):194., J Med Virol. 2023;95(2):e28475., mBio. 2022;13(3): e0130022. Adv Sci. 2023; 10(13):e2207098.). It is scientifically reliable to use SARS-CoV-2 trVLP to evaluate the activity of drugs against SARS-CoV-2. We have added the detailed description of SARS-CoV-2 trVLP/Caco-2-N in the Methods section. And related descriptions in the revised manuscript are as follow:

“SARS-CoV-2 trVLP was kindly provided by Professor Ding Qiang of Tsinghua University and propagated in Caco 2-N cells. In the SARS-CoV-2 trVLP/Caco-2-N system, only the SARS-CoV-2 N gene was replaced by the GFP reporter gene while the engineered Caco-2-N cell line stably expressed and supplemented the viral N protein for live virion package. Previous studies have verified that the SARS-CoV-2 trVLP highly mimicked the transcriptional and replication processes of SARS-CoV-2 and was a live and infectious virus model to test the anti-SARS-CoV-2 efficacy of

drug candidates in a BSL-2 laboratory. SARS-CoV-2 trVLP was not a pseudovirus but a live virus model that was used to test the anti-SARS-CoV-2 abilities of drugs. This model has been widely used in published studies. SARS-CoV-2 trVLP is scientifically reliable model to evaluate the activity of drugs against SARS-CoV-2.” (Line 875, page 37)

Q3. Dauricine

- In Fig. 3E, dauricine at 6.25 uM with full-time showed almost 100% inhibition, although its EC50 was calculated as high as 15.73 uM in Fig. 3A, which is really strange. Same thing happened on daurisoline.

Response: Thanks for your kindly reminder. We have repeated related experiments, and the EC50 and time-of-addition experiments of newly purchased dauricine (Topscience, Cat: T6S0119, Shanghai, China) and daurisoline (Topscience, Cat: T3054, Shanghai, China) against GX_P2V have been re-performed. According to the new experimental data, the EC50 of dauricine against GX_P2V was 13.07 μ M, and the EC50 of daurisoline against GX_P2V was 10.61 μ M. Both dauricine and daurisoline showed almost 100 % inhibition against GX_P2V at 6.25 μ M with full-time (Fig. R1) For the discrepancy between EC50 and time-of-addition results, we think it may be related to different experimental protocols in these two assays. In the EC50 assay, viral loads in the cells are measured 48 h post-infection, whereas in the time-of-addition assay, viral loads in the cells are measured 12 h post-infection. We have described these two assays in detail in the Methods (Line 889, page 37; line 908, page 38).

Fig. R1.

Q4. Isotetrandrone and daurisoline

- Isotetrandrone at 50 uM and daurisoline at 12.5 uM had not big effect on the entry in Fig. 3F. If it is the case, these compounds should not affect on SARS-CoV-2 pseudo typed virus in Fig. 3C, which evaluates SARS-CoV-2 entry. But isotetrandrone and daurisoline showed high anti-entry activity with

EC50 of 7.32 and 8.48 μM , respectively. It is logically unreasonable.

Response: Compared with the inhibitory effect of cepharanthine ($\text{EC}_{50}=0.74 \mu\text{M}$) on SARS-CoV-2 pseudovirus, the inhibitory activity of isotetrandrine ($\text{EC}_{50}=7.32 \mu\text{M}$) and daurisolone ($\text{EC}_{50}=8.48 \mu\text{M}$) were considered to be relatively low. Moreover, in the pseudovirus experiment, the compounds and the virus or the cells are preincubated separately before initiating the infection, and the compounds (eg. isotetrandrine and daurisolone) may partially interfere with the viral entry in this process. This is likely why isotetrandrine at 50 μM and daurisolone at 12.5 μM did not have a major effect on the entry. We have described in detail these two assays in the Methods section (Line 908 and line 953).

Q5. The cytotoxicity of tetrandrine

- As well, the cytotoxicity of tetrandrine was inconsistent between Fig. 3A and 3B.

Response: We have revised the legends of Figure 3 (line 417). Fig. 3A and Fig. 3B contain the cytotoxicity information of tetrandrine in Vero E6 and Caco 2-N cells, respectively. At the same time, the cytotoxicity of tetrandrine in Caco2-N cells was re-analyzed (Fig. R2), and the results were consistent with the previous ones. The same drug shows different cytotoxicity on different cell types, which are reported in several other studies. (Nat Commun 11, 5214 (2020)., Nature 586, 113–119 (2020)., Sig Transduct Target Ther 6, 212 (2021).).

Fig. R2

Q6. Broad spectrum antiviral activity

- Although the authors aim to identify broad spectrum antiviral compounds, they only test on two viruses, GX_P2V and SARS-CoV-2. Only by these data, they cannot claim even for pan-

betacoronavirus or anti-coronavirus activity without SARS-CoV-1 and MERS-CoV. Demonstrating broad spectrum antiviral activity requires infection assay using plus-stranded, minus-stranded, retroviruses, etc. other than coronaviruses.

Response: Thank you for this insightful comment. We have changed the term “broad-spectrum antiviral activity” to “broad-spectrum anti-coronavirus activity” in the manuscript (line 125, 133, 430, 593, 619). In addition, we have supplemented the data on the drug's activities against PEDV and SADS-CoV, as well as SARS-CoV and MERS-CoV pseudoviruses to demonstrate the drug's broad-spectrum anti-coronavirus activity (Figure 4).

Fig. 4 Broad spectrum anti-coronavirus activity assessment of eight BIAs.

Related descriptions in the revised manuscript are as follow:

“We extended our exploration to investigate the broad-spectrum anti-coronavirus activity of these eight BIAs (cepharanthine, tetrandrine, berbamine, fangchinoline, isotetrandrine, cycleanine, dauricine, and daurisoline) against multiple coronaviruses (Fig. 4A-D).

These assays were performed in four coronaviruses: porcine epidemic diarrhea virus (Fig. 4A, EC_{50} = 0.20, 0.44, 0.50, 0.20, 1.67, 0.34, 1.70, and 0.82 μ M, respectively), swine acute diarrhea syndrome coronavirus (Fig. 4B, EC_{50} = 1.03, 2.19, 5.84, 2.23, 3.67, 2.77, 0.84, and 2.80 μ M, respectively), severe acute respiratory syndrome coronavirus (SARS-CoV) pseudotyped virus (Fig. 4C, EC_{50} = 1.37, 3.04, 5.55, 1.73, 4.77, 3.2, 2.63, and 2.76 μ M, respectively), and Middle East respiratory syndrome coronavirus (MERS-CoV) pseudotyped virus (Fig. 4D, EC_{50} = 2.19, 2.17, 3.37, 4.47, 3.31, 2.45, 2.13, and 2.46 μ M, respectively). Intriguingly, these eight BIAs demonstrated generally better inhibitory effects against these coronaviruses than against SARS-CoV-2; an overall comparison between other anti-coronavirus assays and the anti-SARS-CoV-2 assays indicated that the former group exhibited more potent antiviral activity than the latter one (p = 0.03, two-sided t-test). Notably, dauricine outperformed cepharanthine in its efficacy against porcine epidemic diarrhea virus (PEDV) and MERS-CoV (Fig. 4E). Across all seven coronaviruses, cepharanthine consistently demonstrated robust performance, consistently ranking among the top three (Fig. 4F). Daurisoline, dauricine, and cycleanine occasionally secured the first rank. These findings underscore the potential pan-coronavirus inhibitory activity of BIA dimers.” (Line 430-451, page 20)

Q7. Higher antiviral activity than cepharanthine

- The authors' second aim to identify agents having significantly higher antiviral activity than cepharanthine was not achieved either, based on the results shown in Fig. 3C and 3B.

Response: We have modified the expression in the revised manuscript. Among the antiviral assays against seven coronaviruses, we occasionally observed that some cepharanthine analogs have higher antiviral activity than cepharanthine. The text has been revised as follows:

“Notably, cycleanine outperformed, surpassing cepharanthine in GX_P2V- and SARS-CoV-2 trVLP-based assays and ranking as the second-best compound in terms of all three assessments (Fig. 3D). We employed remdesivir, an approved anti-SARS-CoV-2 compound, as a positive control to assess its antiviral activity against GX_P2V and SARS-CoV-2 trVLP (Extended Data Fig. 13).

Remdesivir displayed an EC_{50} value of 1.64 μ M against GX_P2V, which was intermediate between

cepharanthine and cycleanine. For SARS-CoV-2 trVLPs, remdesivir exhibited an EC50 of 0.11 μ M, superior to all tested BIAs.” (Line 402, page 18)

Response to reviewer #3:

The manuscript by Leng et al. described the genome assembly of three *Stephania* species, and evaluated a number of monomeric and dimeric BIA alkaloids produced by these plants for their activity against SARS-Cov2 virus. I find these discoveries provide valuable data for the natural product field to further embark on. Some BIAs showed micro molar range of effective dose EC50 values, indicating that they their structure can be further developed into useful antiviral drugs. The manuscript itself is very well written.

Response: Thanks for your positive and encouraging comments. We have followed your suggestions to improve the manuscript quality.

Q1. The concept of “evo-chemo”

There are several areas that may need substantial improvement for this study:

1) The authors suggested a "evo-chemo" strategy to accelerate drug discovery. At present, there is no support for this claim from this study. Structurally similar compounds are expected to have similar biological functions, known as structure activity relationship. It is possible to evaluate the bioactivities of all the intermediates through the discovery of a biosynthetic pathway. However claiming this as a "evo-chemo" strategy is ambitious, and will need substantial experimental supports from either this study or other studies. So far the authors only have this as a proposal, which did not seem to have accelerated the antiviral drug discovery or be relevant in this manuscript.

Response: Thanks for your constructive suggestions. We followed your suggestion, and deleted confusing expressions of Darwinian medicine, and toned down our conclusion. And related descriptions in the revised manuscript are as follow:

“Main:

“Cepharanthine belongs to benzyloisoquinoline alkaloids (BIAs), a group of nearly 2,500 specialized plant metabolites with remarkable pharmacological effects. Several BIAs are approved pharmacological compounds. For instance, cepharanthine, approved by the Pharmaceuticals and Medical Devices Agency (PMDA) in Japan, is used to treat cancer and inflammation. The well-known antibacterial and hypolipidemic compound berberine, is approved by PMDA and the China Food and Drug Administration (CFDA) to treat infection and parasitology; and tetrandrine has been

approved by CFDA as a calcium channel blocker. Given the potent anti-SARS-CoV-2 property of cepharanthine, the investigation of other antiviral cepharanthine analogs becomes compelling, which inspired us to explore such compounds.” (Line 147, page 6)

Discussion:

“The concept of structure-activity relationship, where structurally similar compounds are expected to share similar pharmaceutical functions, has been employed to identify effective antibacterial drugs. In this study, we propose a methodology to identify new candidate compounds from a known potent antiviral natural product (Fig. 7). Our findings indicate that, similar to cepharanthine, a group of cepharanthine analogs possess potential broad-spectrum anti-coronavirus activity. Furthermore, it is crucial to underscore that this group of analogs stems from the same plant clade as that of cepharanthine. We posit an “adaptive” hypothesis to explain this phenomenon.” (Line 588, page 27)”

Q2. The mode of mechanism

2) While the BIAs showed anti SARS-CoV-2 activity, it is unclear the mode of mechanism. It may be difficult or require a new study to investigate the inhibition mechanism, but at the current stage, the antiviral study lacks a solid conclusion. Not having a clear mechanism also did not help convey the key message of an "evo-chemo" strategy to accelerate drug discovery.

Response: Thank you for your valuable suggestion. The mode of mechanism may involve several possibilities, among which calcium ion (Ca^{2+})-dependent membrane fusion is likely the main mechanism for the inhibitory effects of BIAs on SARS-CoV-2 as well as other coronaviruses (Li et al., 2021, He et al., 2021, Yang et al., 2021). We have discussed the potential mode of mechanism of BIAs against coronavirus infection, which mainly involves BIAs-mediated inhibition on calcium ion (Ca^{2+})-dependent membrane fusion.

Related descriptions in the revised manuscript are as follows:

“Elucidating a well-defined mechanism of action for BIAs against SARS-CoV-2 is crucial for understanding the relationship between structure and antiviral activity, and accelerating drug discovery. The inhibitory effects of bis-BIAs on SARS-CoV-2 and other coronaviruses have been attributed to various mechanisms, with the blockade of calcium ion (Ca^{2+})-dependent membrane fusion by BIAs emerging as a plausible explanation. Bis-BIAs are known as hindering calcium inward flow through the physical alteration of lipid properties, thereby suppressing calcium-induced

responses and inhibiting membrane permeability transition (MPT). This cascade results in inhibiting Ca²⁺-mediated fusion and suppressing virus entry. Variations in the conformation, configuration, and modification of BIA dimers could feasibly impact the physical alteration of lipid properties at varying levels, thereby influencing their antiviral activity.” (Line 505, page 23)

Q3. Biosynthesis of BIA intermediate

3) The authors used the genome assemblies to deduce a number of putative genes for BIA biosynthesis. However none of the putative genes were tested, and they remain putative. If the authors could show the biosynthesis of some BIA intermediate with their putative genes, and combine the anti-viral studies using these intermediates, it may help to substantiate some claims in the "evo-chemo" strategy the authors proposed.

Response: Thanks for your constructive suggestion. We performed enzymatic experiments of norcoclaurine synthase (NCS). The enzymatic experiments on NCS from two syntenic gene array regions were conducted and the genomic basis behind NCS enzymatic activity was discussed. We have redrawn figure 2 and rewritten the corresponding parts:

“We subsequently validated the putative genes for BIA biosynthesis, and all the NCS members from all three species were evaluated, particularly those within two syntenic regions (Fig. 2B, H). These regions encompassed multiple several NCS homologs, forming complete or split gene pairs and arrays. To avoid overlooking possible NCS genes, we conducted local searches for neighboring genes (blast search with E-value = 10⁻¹⁰). The first region comprised seven, seven, and six NCS copies in *S. japonica*, *S. yunnanensis*, and *S. cepharantha*, respectively. The second region contained three, three, and two NCS copies. Catalytic experiments involving 12 NCSs (11 *Stephania* NCSs with high expression and one known-function NCS from *Coptis chinensis* as a positive control) were conducted using dopamine and 4-HPAA as substrates (Fig. 2I). The results demonstrated that five candidate *Stephania* NCS genes possessed NCS functionality. Remarkably, four out of these five *Stephania* NCSs exhibited superior enzymatic activities compared with the *Coptis chinensis* NCS. Notably, these four NCSs were all situated in the second syntenic region (Fig. 2H, I). Phylogenetic analysis unveiled that these four *Stephania* NCSs formed a monophyletic group, with two *S. yunnanensis* NCSs emerging as paralogs due to a recent duplication event (Fig. 2J). These two *S. yunnanensis* NCSs were highly expressed in stems, whereas ScNCS1 and SjNCS4 exhibited high expression in roots. These results underscore the dynamic evolution of *Stephania* NCS genes in

terms of both local copy number adjustments and changes in expression patterns.” (Line 314, page 13)

Fig. 2H-J

Reviewer #4 (Comments to the Author):

In the presented article, titled, “The evo-chemo strategy accelerates anti-SARS-CoV-2 drug discovery”, Leng et al. described a novel strategy to discover new metabolites that could be the potential medicine with broad-spectrum anti-viral properties. Authors reported three plant genomes, one (*Stephania japonica*) being a high quality genome with a single gap, performed some metabolome analysis using these plant species, and then coined the term “evo-chemo” strategy to say the results that they have arrived using comparative genomics and analyzing metabolites for anti-viral properties.

Response: Thanks for your positive comments of our manuscript. We have followed your suggestion to improve the manuscript quality.

Q1. The concept of “evo-chemo”

I feel that the genome assemblies created in this study is good, but that is the only credit I will give to this study. Authors seems absolutely disillusioned, and the study design is everywhere. The results keep jumping everywhere, which is not coherent. Further, I have no idea why authors think that this so called “evo-chemo” is a new strategy, but rather it is just a term that they wish to coin. I do not see any novelty what so ever that may help to discover new drug candidates. The manuscript title is misleading, and since the objective of this study is not clear, authors seems have no clue what they want to report, which makes this study with otherwise nice genome assemblies not impactful. In my opinion, this study is not worth publication at Nature communication in its present form. The datasets are impressive, but the treatment of the datasets are directionless and misleading at occasions in my opinion. Each result section seems incomplete with many questions remaining unexplored. I am listing below reasons, specific comments and my opinion on this study, with the hope that authors find them constructive and could use it for the improvement of this study.

Response: Thanks for your constructive suggestions. We followed your suggestion to change the manuscript title as “Genome-guided Cepharanthine analogs mining in *Stephania* accelerates anti-coronavirus drug discovery”. In accordance with your comments, we have deleted confusing expressions regarding Darwinian medicine and have altered the tone of the conclusion. Related descriptions in the revised manuscript are as shown below:

“Main:

“Cepharanthine belongs to benzylisoquinoline alkaloids (BIAs), a group of nearly 2,500 specialized plant metabolites with remarkable pharmacological effects. Several BIAs are approved pharmacological compounds. For instance, cepharanthine, approved by the Pharmaceuticals and Medical Devices Agency (PMDA) in Japan, is used to treat cancer and inflammation. The well-known antibacterial and hypolipidemic compound berberine, is approved by PMDA and the China Food and Drug Administration (CFDA) to treat infection and parasitology; and tetrandrine has been approved by CFDA as a calcium channel blocker. Given the potent anti-SARS-CoV-2 property of cepharanthine, the investigation of other antiviral cepharanthine analogs becomes compelling, which inspired us to explore such compounds.” (Line 147, page 6)

Discussion:

“The concept of structure-activity relationship, where structurally similar compounds are expected to share similar pharmaceutical functions, has been employed to identify effective antibacterial drugs. In this study, we propose a methodology to identify new candidate compounds from a known potent antiviral natural product (Fig. 7). Our findings indicate that, similar to cepharanthine, a group of cepharanthine analogs possess potential broad-spectrum anti-coronavirus activity. Furthermore, it is crucial to underscore that this group of analogs stems from the same plant clade as that of cepharanthine. We posit an “adaptive” hypothesis to explain this phenomenon.” (Line 588, page 27)”

Q2. The title and the abstract

1. I disagree with the title and the abstract (the last two sentences). The so called evo-chemo strategy is nothing and what people have been using for many many years.

Response: Thanks for your constructive suggestions. We have changed the title to “Genome-guided Cepharanthine analogs mining in *Stephania* accelerates anti-coronavirus drug discovery”. Related expressions in the abstract have also been corrected in the revised manuscript as follows.

“Here, we have systematically assessed anti-coronavirus activity of a series of cepharanthine metabolites from the viewpoint of the biosynthesis pathway, our study will provide an opportunity to accelerate broad-spectrum anti-coronavirus drug discovery.” (Line 122, page 5)

Q3. Fig. 1D

2. Please provide real picture of the plants (Fig 1D). Any specific reason to show roots in the *S. cepharantha*?

Response: We have provided actual pictures of the plants in Figure 1D in the revised manuscript.

Fig. 1D

We did not have any specific reason to show roots in *S. cepharantha*. All *Stephania* plants have conspicuous roots. We have added some high-quality images of the root of *S. yunnanensis* and *S. epigaea* (Figs. R3-4) with the following titles:

Fig. R3. The root of a *S. yunnanensis* plant.

Fig. R4. The root of a *S. epigaea* plant.

Q4. Fig. 1B and karyotyping data

3. Please provide chromosome names that has been used for specific plant genomes to the contact maps (Fig 1B). In figure 2B, authors show chromosome 1 of *S. japonica* having relationships with two chromosomes of *Syun* and *Scap*. What is chromosome 1 of *Sjap* in the contact map? In the contact map, fifth block from the bottom for *S. japonica* shows some strange structure, please comment on that. Also, please The box boundary is missing for third box for *S. japonica* and 7th box for *S. yunnanensis* in Fig 1B. Also, Please provide karyotyping data here. I think that this data will ensure that the chromosome numbers matches with pseudo molecules reported in this study. Even a reference of chromosome numbers will be good enough. Further, *S. japonica* has 11 chromosomes, two of which (chromosome 1 and chromosome 2) seems to have associated with four chromosomes of *Syun* and *Scep*, resulting in 13 chromosomes. So, what happens first? I mean, the two chromosomes of *Sjap* resulted in four chromosomes of *Syun* and *Scep*, or the fussion of two chromosomes resulted in a single chromosome of *S. japonica*. Please consider talking about this interesting aspect of the genomes.

Response: Thanks for your constructive suggestions. We have followed your constructive suggestions to improve the quality of our manuscript. And we have added chromosome names to the contact maps (Fig. 1B). We have also fixed the box boundary for the third box for *S. japonica* and the 7th box for *S. yunnanensis*.

Fig. 1B.

We also extracted the HiC contact information of *Sjap* chromosome 1 and generated the contact map using HiCExplorer (Wolff et al., 2018). The HiC contact map of *Sjap* chromosome 1 supports the

integrity of this chromosome (Fig. R5).

Fig. R5. The contact map of *S. japonica* chromosome 1.

The fifth block from the bottom for *S. japonica* shows some strange structure (Fig. R6). This block represents chromosome 7. Two regions in this chromosome show questionable contact signals, which usually imply local mis-assembly. However, we carefully checked the assembled sequences both in the contig level or after HiC-assistant scaffolding, but we found no evidence of mis-assembly. The alternative explanation is that these two regions enrich repetitive sequences, e.g., one is the centromere region while the other enriches repetitive sequences similar to centromere region repeats. We thus employed the HiCExplorer pipeline to reanalyze the HiC sequencing data. Compared with the ALLHIC pipeline, the HiCExplorer pipeline is more stringent regarding read mapping and uniquely mapped read selection. More than 1,009 million reads were kept for visualization using the ALLHIC pipeline, while ~235 million reads were kept using the HiCExplorer pipeline. The HiCExplorer pipeline generated a much more reasonable result for chromosome 7 (Fig. R7). The left region (highlighted in a blue square, Fig. R6) no longer shows an abnormal signal, while the right region (highlighted in a yellow square, Fig. R6) currently looks like a centromere region. The coordinate of this putative centromere region's left boundary is roughly 30,160,000 bp, while the right is 31,210,000 bp. The TRF (tandem repeat finder) result and gene density distribution suggest this region is within the scope of the predicted centromere and peri-centromere of chromosome 7 (Fig. R8). Lastly, we mapped ONT ultra-long reads to this region and checked whether ultra-long reads could be successfully mapped (Fig. R9). Both boundaries and all internal parts of the focal region were covered with ultra-long reads. These lines of evidence verified that the problematic

region in Sjac chromosome 7 is unlikely to be mis-assembly.

Fig. R6. Putative problematic regions in Sjac chromosome 7. The blue and yellow squares highlight two regions that may result from mis-assembly.

Fig. R7. Redrawn contact map of Sjac chromosome 7.

Fig. R8. Gene density and repeat (TRF) density around predicted centromere and peri-centromere region of Sjak chromosome 7.

Fig. R9. Mapping of ONT ultra-long reads on chromosome 7: 30,160,000-31,210,000. Each read was denoted by a line, with its color randomly assigned to distinguish it from its neighbors. A and B showed the left and right boundary of the focal region, while C to F showed four randomly picked blocks within the focal region.

We retrieved chromosome numbers of *Stephania* species from the chromosome counts database (CCDB, <http://ccdb.tau.ac.il>). From the CCDB data, there are three possible chromosome numbers of the *Stephania* species (Fig. R10). Among the three *S. subgenus* *Stephania* species, *S. japonica* and *S.*

elegans have 11 chromosomes, while *S. hernandifolia* has 12. Among the three *S. subgenus* *Tuberiphania*, *S. cepharanthine* has 11 chromosomes, while *S. yunnanensis* and *S. dielsiana* have 13. The three species we chose for sequence were *S. japonica* (11 chromosomes according to CCDB), *S. yunnanensis* (13 chromosomes), and *S. cepharantha* (11 chromosomes). However, according to our HiC-based pseudo-chromosome construction, we found that both *S. yunnanensis* and *S. cepharantha* have 13 chromosomes, while *S. japonica* has 11. It is now difficult to determine the exact chromosome number of *Stephania* species. Our data support the possibility that species of *S. subgenus* *Tuberiphania* have 13 chromosomes while species of *S. subgenus* *Tuberiphania* have 11 or 12 chromosomes. We further searched chromosome numbers of *Cyclea* and *Cissampelos*, two genera closely related to *Stephania*. Interestingly, *Cyclea racemosa* has 12 chromosomes, and *Cyclea peltate* has 24, but both *Cissampelos glaberrima* and *Cissampelos pareira* have 12. These results indicate that the ancestor of all existing *Stephania* species might have 12 chromosomes, and the chromosome number has been evolving to 11, 12, and 13. So, the exact karyotype evolution history of *Stephania* remains to be determined in the future.

We could perform karyotype evolution analysis (e.g., doi.org/10.1093/hr/uhad139), and we sincerely thank the reviewer to pinpoint that *Stephania* karyotype evolution history contains interesting aspects of the genomes. As we have already had the nearly T-2-T level assembly of *S. japonica*, we are working to resolve the T-2-T assembly of *S. cepharantha*. If we are successful, we will discuss in detail about the karyotype evolution history in a subsequent study.

Fig. R10. Karyotyping data of *Stephania* species retrieved from the chromosome counts database (CCDB, <http://ccdb.tau.ac.il>). The *S. subgenus* *Stephania* species are marked with a bright blue background, *S. subgenus* *Stephania* species with bright orange, and *S. subgenus* *Stephania* species with bright green, respectively. The phylogenetic relationship of all *Stephania* species was constructed according to Bayesian analyses of the ITS sequences (Xie et al., 2015).

Q5. Information about the experimental design in the introduction

4. Please provide more information about the experimental design in the introduction. I mean, why you choose *S. jap*, *S. yun* and *S. cep*? Three plants from the same genus and same phylum to do what? Why not choose plants from same genus that are not producing the metabolites with medicinal properties. In that case, the resource will help in answering a lot of interesting questions. In this study, authors would have concluded exactly the same thing using *S. japonica* genome assembly.

Response: Thanks for your kindly reminder. When we planned to study the analogs of cepharanthine, we searched a large number of literatures to summarize the phytochemistry informations of *Stephania* genus. Unfortunately, we did not have any knowledge about which species from the same genus do not produce cepharanthine. Therefore, we chose three species from the same genus to ensure we could successfully assemble at least one *Stephania* genome and find metabolites with and without antiviral activity. We found that BIA monomers generally do not have antiviral activity,

while many BIA dimmers do.

Q6. More information about the cepharanthine and intermediates

5. Please provide more information about the cepharanthine and intermediates, what is known, what is not known and so on. Any FDA approved drug for this molecule or its derivative? What plant families produces this compound. Does intermediates are previously reported? In the abstract section, authors mentioned about seven of nine cepharantine analogs with anti-coronavirus properties, so these metabolites were not know before? These are the new metabolites reported for the first time ever? This aspect is completely unclear. Authors need to provide more details as where these analogs have previously reported and so on. Authors mentioned about *Nelumbo* and *Thalictrum*, and that they produces ceptharanthine analogs, so, these are different from *Stephania*? How these are different? Also, why authors opted to sequence three genomes from *Stephania* but not from *Nekumbo* and *Thalictrum*. Experimental design wise, include plants from these families would have provide much important insights on the biosynthesis pathways. This is a major weekness.

Response: Thank you for pointing this out. We have added more information about the cepharanthine and intermediates as you suggest. And none of thes compounds is approved by the FDA, but some were approved by CFDA (China) or PMDA (Japan). These intermediates were previously reported, but little is known about their antiviral activity. Except for cepharanthine, knowledge of the analogs' anti-coronavirus activity was scattered in multiple studies. To our knowledge, this work is the first study that systematically investigates the anti-coronavirus activity of cepharanthine analogs, especially from the viewpoint of biosynthesis pathway. No new metabolite was reported for the first time, because if a new metabolite was discovered, it is not easy to sufficiently collect sufficient such compound for various antiviral experiments. The two genera *Nelumbo* and *Thalictrum* could produce BIA dimmers, but these BIA dimmers differ from *Stephania*'s. The *Nelumbo* genome has been sequenced and assembled several times, but the *Thalictrum* genome is to be assembled. We will sequence the *Thalictrum* genome in the future. Related descriptions in the revised manuscript are as follows:

“Cepharanthine belongs to benzylisoquinoline alkaloids (BIAs), a group of nearly 2,500 specialized plant metabolites with remarkable pharmacological effect. Several BIAs are approved pharmacological compounds. For instance, cepharanthine, approved by the Pharmaceuticals and

Medical Devices Agency (PMDA) in Japan, is used to treat cancer and inflammation. The well-known antibacterial and hypolipidemic compound berberin, is approved by PMDA and the China Food and Drug Administration (CFDA) to treat infection and parasitology; and tetrandrine has been approved by CFDA as a calcium channel blocker. Given the potent anti-SARS-CoV-2 property of cepharanthine, the investigation of other antiviral cepharanthine analogs becomes compelling, which inspired us to explore such compounds.

In this study, we assembled high-quality *Stephania* genomes, proposed a cepharanthine biosynthesis pathway, systematically investigated the anti-coronavirus activity of metabolites along this pathway, and deliberated on avenues for leveraging chemodiversity to advance drug discovery.” (Line 147, page 6)

“Considering that *Stephania* yields a cluster of antiviral compounds, we were intrigued to determine whether the antiviral activity was unique to the *Stephania* clade or extended to a broader range of plants producing bioactive cepharanthine analogs. BIAs are largely restricted to certain plant families primarily in the order Ranunculales. Consequently, BIA metabolism research has mainly focused on plants from the families *Papaveraceae*, *Menispermaceae*, *Ranunculaceae*, and *Berberidaceae*. Among them, cepharanthine and tetrandrine, derived from the *Menispermaceae* family, have demonstrated anti-SARS-CoV-2 activity. Additionally, BIAs are sporadically found in the orders Piperales and Magnoliales as well as in the families *Rutaceae*, *Lauraceae*, *Cornaceae*, and *Nelumbonaceae*. To date, BIAs have not been identified in gymnosperms, monocots, or Amborellales. We performed a search to identify eudicot plants producing BIAs with reported potential antiviral activity. Neferine and liensinine, major bisbenzylisoquinoline components and bioactive constituents of sacred lotus (*Nelumbo nucifera*), have shown potential as SARS-CoV-2 inhibitors. Sacred lotus, a basal eudicot plant, produces numerous BIAs. For our study, we selected the genus *Thalictrum* among Ranunculales plants, considering its taxonomic proximity to BIA-producing plants. Our search yielded five antiviral analogs from *Nelumbo* and *Thalictrum* (Fig. 6A): liensinine and neferine from *Nelumbo*, and thalidezine, thalmineline, and methoxyadiantifoline from *Thalictrum*. No BIAs have been reported in gymnosperms. Moreover, Amborellales, early-diverging angiosperms, do not accumulate these compounds. Consequently, BIA biosynthesis appears to have evolved after the divergence of Amborellales and other angiosperms but before the separation of magnoliids, eudicots, and monocots (Fig. 6B). Moreover, certain compounds with anti-SARS-CoV-2

activity originate from several early-diverging eudicots such as *Nelumbo* from Proteales and *Thalictrum* and *Stephania* from Ranunculales. Two explanations may account for the phylogenetic distribution of plants that produce and accumulate antiviral cepharanthine analogs: first, the activity being evolved in three lineages independently, and second, the activity emerging in the early history of angiosperm evolution but being lost subsequently in some lineages.” (Line 534-563, page 25)

Q7. The objective of this study

6. In the introduction, authors mention objective in page 6, from line 148 till line 151. I disagree as genome sequencing and comparative genome analysis has been the way forward to answer exactly this question for several years. In these previous studies, intermediates have been identified across multiple plant species, and comparative genome analysis have explained the emergence of specialized metabolites. Genome studies of Opium poppy, *Camptotheca acuminata*, *Ophiorrhiza pumila*, and so many other medicinal plant species are example. So, the objective is not really an objective, rather misleading.

Response: We apologize for this inadvertent oversight. The related descriptions in the revised manuscript are as follows:

“In this study, we assembled high-quality *Stephania* genomes, proposed a cepharanthine biosynthesis pathway, systematically investigated the anti-coronavirus activity of metabolites along this pathway, and deliberated on avenues for leveraging chemodiversity to advance drug discovery.” (Line 157, page 6)

Q8. Single-copy gene

7. Line 181, “...species and constructed.....”, what is strictly single-copy gene. Single copy gene is single copy gene. This is not clear. Please clarify.

Response: Thank you for pointing this out. We have deleted the unnecessary word “strictly”. (Line 187, page 7)

Q9. Definition of the term “*Stephania*”

8. Line number 184, authors says that *S. japonica* belongs to the subgenus *Stephania*, and the rest of

the two plants belong to subgenus *Tuberiphania*, then why authors keep using *Stephania* for all the three plant genomes. It does not make sense and very confusing. I am not sure what classification authors are using. Authors have used *Stephania* to describe the plant species together, but then if they belong to different subgenus, then why to use this way?

Response: Related descriptions in the revised manuscript are as follows:

“The estimated phylogeny and divergence timings are consistent with *S. japonica* belonging to the *S.* subgenus *Stephania* (because the genus and the subgenus are both named “*Stephania*”, the term “*Stephania*” exclusively represents the genus hereinafter for clarity), while *S. cepharantha* and *S. yunnanensis* belong to the *S.* subgenus *Tuberiphania* (Fig. 1A).” (Line 189, page 8)

Q10. WGD analysis

9. Line 203 to line 207, authors mention that *S. japonica* has not undergone shared whole genome triplication that is shared between core eudicots, and therefore, they reported 2:3 synteny ratio between *S. japonica* and *V. vinifera*. This is reasonable. But then how authors can say that *Stephania* species diverged from *V. vinifera*? The synonymous substitution plot (Fig 1E) is very hard to understand. Please consider choosing the color properly, and if possible or like, separate the orthologs and paralogs sets.

Response: Related descriptions in the revised manuscript are as follows:

“Another WGD event in *Stephania* species occurred after the divergence of its ancestor from *V. vinifera* (Fig. 1F, Extended Data Fig. 6)”. (Line 214, page8)

We have modified Fig. 1E accordingly.

Fig. 1E

Q11. Gap free

10. Line 198, it says that *S. japonica* genome assembly is gap free, but it has a single gap. Please correct this point.

Response: We apologize for this oversight. We have now deleted this term, and related descriptions in the revised manuscript are as follows:

“...resulting in a near-telomere-to-telomere assembly for *S. japonica* genome and chromosome-level assembly of the *S. cepharantha* and *S. yunnanensis* genomes (Fig. 1C, D, Extended Data Fig. 5).”
(Line 205, page 8)

Q12. Genes and their sequences

11. Unless I have missed, please provide a supplementary table with genes that were used to identify homologs for NCS, methyltransferases, and cyp450. Also, if possible, create a github repository for this manuscript, and provide the sequences used here. Further, in line 252, authors mention about 267, 331 and 234 potential enzymes, so, these numbers of NCS, methyltransferases and cyp450 combined for a specific plant species.

Response: Thanks for your helpful suggestions. We have added several tables and .fasta format files to store all information and sequence of genes that were used to identify homologs for NCS, methyltransferases, and cyp450, and generated a GitHub repository for this manuscript. These files are quite large, thus we put them on the github repository (https://github.com/liuzy2008/evo-chemo_anti-SARS-CoV-2_drug_discovery2023). The corresponding part in “Data and materials

availability” now reads as:

“The information of functional BIAs-biosynthetic genes is hosted in GitHub (https://github.com/liuzy2008/evo-chemo_anti-SARS-CoV-2_drug_discovery2023).” (Line 1005, page 41)

Q13. Additional analysis of BCGs in Syun

12. Authors describe BCGs from Line 260 onwards. Here, authors talk about the expression of genes that are conserved and part of BCGs. But I am not sure why authors describe Sjav and Scep, and not Syun? In principle, the three genomes are highly conserved, in which, Scep and Syun is closer and have higher synteny compared with Sjav. So, how many BCGs are conserved across all these three species? Is expression is consistent between Scep and Syun? What tissue is well known to produce the key medicinally relevant metabolite? Does the expression is consistent in BCGs for tissues which produces high level of key metabolites? Does the tissue that accumulated the key medicinal BIAs are same for all the plant genomes? These are basic questions that is completely ignored which other wise would provide interesting aspects of BIA dimer biosynthesis. The comparative genome analysis between these three plants are ignored, and at times, not consistent. Like, in this case, the comparison is described between Sjav and Scep, but Syun is completely ignored, and as a reader, I wonder why? Authors talk about co-inheritance between two species, but what happens when Syun is included? “incompletely consistent” is not the proper way to mention.

Response: Thank you for these insightful comments. We only described Sjav and Scep due to space limitation, especially figure 2. Besides, this study concentrates on the anti-coronavirus activity of BIAs; the analysis of BGCs is a part that could not be ignored in the whole story, but not the most important one. Thus, we only put the comparison between *S. cepharantha* and *S. japonica*, and added the corresponding analysis of *S. japonica/S. cepharantha* versus *S. yunnanensis* in the Extended Data Fig. 9. Related descriptions in the revised manuscript are as follow:

“When comparing the BGCs between *S. cepharantha* vs. *S. yunnanensis* and *S. japonica* vs. *S. yunnanensis*, the same pattern could be observed (Extended Data Fig. 9).” (Line 295, page 13)

According to Figure 2B, nearly all BGCs are conserved across all these three species. However, this conclusion depends on how we define a BGC (PMID: 35441651). In this study we avoided discussing the whole genomic picture of BGCs, but took two cases as example (Fig. 2 C-F, Extended Data Fig. 9). Generally, the key metabolites are mainly accumulated in root (Fig. 2G). However,

since we do not know the exact enzymatic activity of cluster genes, e.g., substrate and product of these OMTs and CYPs (Fig. 2C-E, Extended Data Fig. 9 A-D, F), we could hardly draw a conclusion that the expression of putative enzyme genes is consistent in BCGs for tissues which produces high level of key metabolites. The sentence was modified to “However, gene expression patterns within the gene cluster in *S. japonica* do not entirely align between the two species.” (Line 273, page 12)

Extended Data Fig. 9].

Q14. Graph based pangenome

13. I recommend authors to consider drawing a graph based pangenome for the *Stepmania* species. I understand that this is only three genome, but it will be nicer to see as what constitutes core genome,

what is unique to each species, and if some functional aspect comes up through this. Since the genome is nice, authors could even identify the structural variants within individual genomes, and then see the similarity and differences that exists in terms of the distribution of these variants across the three plant genomes described here.

Response: Thanks for your constructive suggestions. We tried to perform the graph based pangenome analysis using several tools. Minigraph-cactus (Hickey et al., 2023) and nucmer/svmu (Huang et al., 2023) did not generate a meaningful result. Although the SYRI pipeline could generate some result, an in-depth analysis of the SYRI results revealed that applying the pangenome analysis method on three *Stephania* species, especially *S. japonica* and other two species with divergence time being ~20Mya, was problematic. For example, the ‘core’ part from SYRI analysis is only ~62 Mb, making up for less than 10% of any species’ genome (Fig. R11). This counter-intuitive phenomenon implies that pangenome analysis may not fit these three species.

Fig. R11. Venn plot of overlapping regions between *S. japonica*, *S. cepharantha*, and *S. yunnanensis*.

Q15. BIA biosynthesis pathway

14. Authors mentioned that several BIAs from *Stephanias* had been pandreviously reported. Any new metabolites that authors detected in this study? If not, then these metabolites were already known, right? How authors confirmed the metabolites identity, using standards or MS/MS analysis? How authors excluded the false positives? The proposed pathway for BIAs in *Stephania* species were drawn based on what? The identified metabolites chemical structure were considered to place them in the sequence or the order of the reaction is already know? Again, authors keep talking about cepharanthine biosynthesis pathways, but at occasions, it feels like the pathway is known and at

occasions, it feels like this study is proposing the pathway. It is very confusing and authors need to clarify this aspect.

Response: These intermediates were previously reported but knowledge of their antiviral activity was limited. Like cepharanthine, knowledge of the analogs' anti-coronavirus activity was scattered in multiple studies. To our knowledge, this work is the first study to systematically investigated the anti-coronavirus activity of cepharanthine analogs, especially from the biosynthesis pathway's point of view. All the cepharanthine analogs in biosynthetic pathway were already known. We identified metabolites using standards combined with MS/MS analysis (Supplementary Table 9). We have added the manufacturer information for these standard products in Supplementary Table 9. And other non-commercial compounds were identified based on our own MS/MS analysis. To exclude false positive results, we also referred to the secondary mass spectrogram of these compounds in other literature. We proposed the pathway for BIA in *Stephania* species based on their scaffold, modification, and possible enzymes, such as OMT or P450 that could use one identified metabolite as the substrate to produce another metabolite. We have modified several sentences to clarify that the pathway is proposed, and related descriptions in the revised manuscript are as follows:

“Here, we have systematically assessed anti-coronavirus activity of a series of cepharanthine metabolites from the viewpoint of biosynthesis pathway, our study will provide an opportunity to accelerate broad-spectrum anti-coronavirus drug discovery.” (Line 122, page 5)

“Proposal of the putative cepharanthine biosynthesis pathway” (Line 236, page 11)

Q16. The positive control in anti-viral experiment

15. For the anti-viral experiment, what is the positive control, daurisoline? Tetrandrine is control for cytotoxicity? Anti-viral activity of cycleanine is never reported before and this is the first time that it was detected? Authors says that the BIA dimers showed paninhibitory activity of BIA dimers, which was also the previous observation. Then what is the novelty of the entire section (Line 331 till 367). Authors could have started this section with line number 368. I am not sure what authors then identified here in this section. Dauricine is used molecule as anti-viral drug?

Response: We have added remdesivir as the positive control in antiviral experiments (Extended Data Fig. 13). Each BIA has been tested for EC50 (indicator for antiviral activity) and CC50 (indicator for

cytotoxicity), and tetrandrine is not a control for cytotoxicity here. Moreover, to our knowledge, the antiviral activity of cycleanine has never reported and this is the first time that it has been detected. Besides cepharanthine, knowledge of the analogs' anti-coronavirus activity was scattered in multiple studies. This work is the first study that systematically investigates the anti-coronavirus activity of cepharanthine analogs and BIAs, especially from the viewpoint of biosynthesis pathway. Dauricine is one of the BIAs we found in this study with potent anti-coronavirus activity.

Related descriptions in the revised manuscript are as follows:

“We employed remdesivir, an approved anti-SARS-CoV-2 compound, as a positive control to assess its antiviral activity against GX_P2V and SARS-CoV-2 trVLP (Extended Data Fig. 13). Remdesivir displayed an EC₅₀ value of 1.64 μM against GX_P2V, which was intermediate between cepharanthine and cycleanine. For SARS-CoV-2 trVLPs, remdesivir exhibited an EC₅₀ of 0.11 μM, superior to all tested BIAs.” (Line 401, page 18)

Extended Data Fig. 13].

Q17. Structural changes between Stephania BIA dimers and antiviral activity

16. The entire section starting from line 388 is based on assumptions and not fact. Authors could have tested if the BIA dimers have different affinity with the target proteins compared with the monomers through docking. Further, why authors assumed that the dimerization is the decisive step

towards anti-viral properties? What is the basis? Previous studies claimed it? If yes, the describe it, and the refute the claim through your data. Authors does talk about putative genes but they do not test it to confirm, and then this analysis or even discussion becomes meaningless.

Response: Thanks for your valuable suggestion. We have followed your suggestions and performed molecular docking to compare the affinity of the BIA dimer and the monomer to the target protein. We have added the relevant description in the main text as follows:

“Molecular docking calculations provided similar results (Fig. 5B, C). Notably, the binding energy between BIA dimers and the SARS-CoV-2 spike protein-ACE2 complex (7VX4 or 7VX5) was significantly higher than that of BIA monomers ($p < 0.0001$, two-sided Wilcoxon test, Fig. 5B). However, no clear correlation existed between antiviral activity and binding energy of BIA dimers (Fig. 5C). Furthermore, no evidence demonstrated that the conformation or modification of BIA dimers influenced their antiviral activity.” (Line 498, page 23)

Fig. 5B-C

We speculated that dimerization is a key step for the antiviral activity of BIAs, but not the only one. We also performed enzymatic experiments of norcochlorine synthase (NCS). The enzymatic experiments on NCS from two syntenic gene array regions were conducted and the genomic basis behind NCS enzymatic activity was discussed. We have redrawn figure 2 and rewritten the corresponding parts, and related descriptions in the revised manuscript are as follow:

“We subsequently validated the putative genes for BIA biosynthesis, and all the NCS members from all three species were evaluated, particularly those within two syntenic regions (Fig. 2B, H). These regions encompassed multiple several NCS homologs, forming complete or split gene pairs and arrays. To avoid overlooking possible NCS genes, we conducted local searches for neighboring genes (blast search with E-value = 10^{-10}). The first region comprised seven, seven, and six NCS copies in *S. japonica*, *S. yunnanensis*, and *S. cepharantha*, respectively. The second region contained

three, three, and two NCS copies. Catalytic experiments involving 12 NCSs (11 *Stephania* NCSs with high expression and one known-function NCS from *Coptis chinensis* as a positive control) were conducted using dopamine and 4-HPAA as substrates (Fig. 2I). The results demonstrated that five candidate *Stephania* NCS genes possessed NCS functionality. Remarkably, four out of these five *Stephania* NCSs exhibited superior enzymatic activities compared with the *Coptis chinensis* NCS. Notably, these four NCSs were all situated in the second syntenic region (Fig. 2H, I). Phylogenetic analysis unveiled that these four *Stephania* NCSs formed a monophyletic group, with two *S. yunnanensis* NCSs emerging as paralogs due to a recent duplication event (Fig. 2J). These two *S. yunnanensis* NCSs were highly expressed in stems, whereas ScNCS1 and SjNCS4 exhibited high expression in roots. These results underscore the dynamic evolution of *Stephania* NCS genes in terms of both local copy number adjustments and changes in expression patterns.” (Line 314, page 14)

Fig. 2H-J

Q18. Other genera with antiviral compound discovery potential

17. Section starting from Line 431 seems to me clueless. Why authors choose these plants? Why not perform genome sequencing for selected plants from these different genus and not all three from the same? Authors say that they found five antiviral analogs, from where, gymnosperms, monocots or Amborellales? The placing of this sentence is misleading and I am not able to understand purpose

here. Authors says that no BIAs have been reported in gymnosperm, then please use some genomes from gymnosperm to explore biosynthesis pathways for BIAs dimers that authors are describing here. Also, how authors can make this claim? Any reference?

Response: Thanks for your kindly reminder, and we are sorry for the misleading. BIAs are mainly distributed into early-diverging eudicots, but there are not any reports in gymnosperms, monocots and Amborellales. We have downloaded all published genomes from gymnosperms, monocots and Amborellales to identify the BIA biosynthetic genes. The results showed that the crucial NCS, CYP719 and CYP80 genes related to BIA biosynthesis were missing, was consistent with the BIA distribution. And we have revised this part of the content to enhance its readability and accuracy. The revised content is as follows:

“Neferine and liensinine, major bisbenzylisoquinoline components and bioactive constituents of sacred lotus (*Nelumbo nucifera*), have shown potential as SARS-CoV-2 inhibitors. Sacred lotus, a basal eudicot plant, produces numerous BIAs. For our study, we selected the genus *Thalictrum* among Ranunculales plants, considering its taxonomic proximity to BIA-producing plants.” (Line 546, page 25)

Q19. The concept of “evo-chemo”

18. I completely disagree with entire section starting from line no 466. This is not a new strategy, and has been in practice since always. Did authors elucidated the pathway here? Authors in principle says that you need to know anti-viral compound, its properties and which plant produces it. Then, analyze the plant, find the intermediates, then then start looking for new metabolites that could have anti-viral activities. But the choice of metabolite is because that was already tested and clinically proven. Then what is the point. You can predict the improved ligand binding through molecular docking, and can guess even as what changes in the chemical can result in better action.

Response: We appreciate your constructive suggestions. Accordingly, we have modified the expression related with concept of “evo-chemo” in the revised manuscript as follows:

“Main

“Cepharanthine belongs to benzylisoquinoline alkaloids (BIAs), a group of nearly 2,500 specialized plant metabolites with remarkable pharmacological effects. Several BIAs are approved pharmacological compounds. For instance, cepharanthine, approved by the Pharmaceuticals and

Medical Devices Agency (PMDA) in Japan, is used to treat cancer and inflammation. The well-known antibacterial and hypolipidemic compound berberine, is approved by PMDA and the China Food and Drug Administration (CFDA) to treat infection and parasitology; and tetrandrine has been approved by CFDA as a calcium channel blocker. Given the potent anti-SARS-CoV-2 property of cepharanthine, the investigation of other antiviral cepharanthine analogs becomes compelling, which inspired us to explore such compounds.” (Line 147, page 6)

Discussion:

“The concept of structure-activity relationship, where structurally similar compounds are expected to share similar pharmaceutical functions, has been employed to identify effective antibacterial drugs. In this study, we propose a methodology to identify new candidate compounds from a known potent antiviral natural product (Fig. 7). Our findings indicate that, similar to cepharanthine, a group of cepharanthine analogs possess potential broad-spectrum anti-coronavirus activity. Furthermore, it is crucial to underscore that this group of analogs stems from the same plant clade as that of cepharanthine. We posit an “adaptive” hypothesis to explain this phenomenon.” (Line 588, page 27)

We have also predicted improved ligand binding through molecular docking, related description in the revised manuscript are as follow:

“Molecular docking calculations provided similar results (Fig. 5B, C). Notably, the binding energy between BIA dimers and the SARS-CoV-2 spike protein-ACE2 complex (7VX4 or 7VX5) was significantly higher than that of BIA monomers ($p < 0.0001$, two-sided Wilcoxon test, Fig. 5B). However, no clear correlation existed between antiviral activity and binding energy of BIA dimers (Fig. 5C). Furthermore, no evidence demonstrated that the conformation or modification of BIA dimers influenced their antiviral activity.” (Line 498, page 23)”

Figure 5 B-C.

Q20. Statements in discussion

19. I am now sure what is the basis for the authors to claim that “ Our result showed that the chemo....(Line 509)”. I do not see any result leading to this statement in this article. Similarly, no

results is presented to claim line 544, “Our results laid the foundation...”. Authors reported genomes, and metabolites were previously reported. What else?

Response: We have modified this sentence in the revised manuscript as follow:

“This study is the first to assemble high-quality *Stephania* genomes, and examine the anti-coronavirus activity of cepharanthine analogs and BIAs from a biosynthetic pathway perspective. Undoubtedly, these findings will expedite the search for broad-spectrum anti-coronavirus.” (Line 618, page 28)

Q21. Methods of metabolomics

20. Line 707, “a UPLC-MS/MS system...” seems wrong. Please confirm. Is this a triple Q ion trap or Triple TOF? These are two different things. Are they in-line? Why? What is the advantage and how it helped them? What fragments (daughter ion) for a specific metabolites were used for MRM analysis?

Response: Thank you for pointing this out. We used Triple TOF to detect the alkaloid content of the samples. The TOF-MS system has a high resolution, which helps to reduce the interference of m/z approximate ions and enables a more comprehensive qualitative analysis of low abundance analytes in complex materials. It can detect many compounds in complex samples and help distinguish compounds with similar molecular weights. We used SCAN mode, not the MRM mode, for ion screening to ensure that effective information should not be missed. We apologize for our error and appreciate your highlighting of this issue. Related descriptions in the revised manuscript are as follows:

“To investigate the content of BIAs in different tissues of the three *Stephania* species, a ultra-performance liquid chromatography–tandem mass spectrometry system (SCIEX TripleTOF 6600+), ultra-high–performance liquid chromatography (ultra-HPLC) coupled with time-of-flight mass spectrometry was used for the relative quantification of metabolites.” (Line 794, page 34)

“The mass spectrometer was used in the full scan mode at a scan time of 35 ms per transition.” (Line 808, page 35)

Reviewers' Comments:

Reviewer #1:

Remarks to the Author:

As Reviewer1, I am glad that the authors have tone-downed the use of "eco-chemo" strategy in search of antiviral compounds.

For Q2, why only the NCS was analyzed? are there other diversified loci among these genomes contributing to the chemical diversity?

Reviewer #2:

Remarks to the Author:

The authors did not respond to the reviewer's comments, especially the infection experiment using SARS-CoV-2 live virus instead of VLP (Q2) that is essential for supporting the authors' conclusion. Experimental results obtained with VLP has to be confirmed using SARS-CoV-2 live virus since the VLP is an artificial system as the reviewer commented, but the authors did not answer to the comment. As well, the authors did not improve the description of manuscript in which they still claim "broad-spectrum anti-coronavirus activity" without any experiment using SARS-CoV-1 and MERS-CoV, neglecting the reviewer's comment (Q6). They did not sufficiently improve the inconsistent results between figures (Q4, 5).

The manuscript was not sufficiently improved to be able to be published in the journal.

Reviewer #3:

Remarks to the Author:

The revised manuscript has addressed a few minor concerns of mine, including adding discussion on BIA inhibition mechanism and performing NCS assays, as well as the new NCS gene duplication discovery. NCS is the well-known gateway enzyme for BIA. I was hoping that the authors could make new compounds with the putative genes from the genome, to show the use of it. The authors also removed their claim on evo-chemo strategy for accelerating drug discovery.

The two individual components of this manuscript, namely 1) genome assembly and small discovery of NCS duplication, 2) anti-SARS-Cov2 assays with cephalanthine/derivatives etc. are separately sound, and separately useful to researchers in two different fields. Yet the authors tried hard to promote the idea that the genome assembly has guided and been useful for their discovery of other anti-SARS-cov2 compounds. Still there has been no support from this manuscript on this matter, and it remains strange to marry this two components in this manuscript. Clearly the genome mining has not accelerated anti-SARS-Cov2 drug discovery in this article, nor has the authors created new, superior intermediates or derivatives using their new genes, benefiting from the genome assembly.

Publication of this work requires substantial new data to show that the genome mining has led to the discovery of new genes, which have led to the biosynthesis of unknown or undetected BIAs superior to cephalanthine. Alternatively, the author may consider separating the two components in two articles.

Reviewer #4:

Remarks to the Author:

I am pleased to see that the authors have addressed all my concerns. I am still not confident about the genome assembly's accuracy based on viewing the HiC contact map. The authors provided enough evidence to say that these are likely not assembly errors when explaining chr1 of *S. japonica*. But then, when I check their HiC contact map for *S. yunnanensis*, I can see the 3rd box telomere

region(from the top) has strange contact with 9th box putative centromere region. If I had this assembly, I would correct this issue, cut this region, and then move it down to 9th box as most likely, this is repeats, but telomeric and centromere regions should be quite distinct. As I mentioned in my previous comments, this aspect requires more careful validation for each chromosome. Once a genome assembly is established, there is little incentive to correct it or validate its correction, and therefore, I feel that it is important to check each chromosome for a genome assembly before it is made public. In my opinion, This remains an issue for me. Other than that, authors have done a great job of improving the text and added more information. Lastly, I request authors to please include all your scripts that were used for creating figures, analysis, TAD analysis, gene cluster analysis and so on. This is important for open science, and I am not able to find scripts used for genome assembly and comparative genome analysis in the mentioned github.

Responses to

Reviewer #1 (Comments to the Author):

As Reviewer1, I am glad that the authors have tone-downed the use of "eco-chemo" strategy in search of antiviral compounds.

For Q2, why only the NCS was analyzed? are there other diversified loci among these genomes contributing to the chemical diversity?

Response: We are deeply thankful for your constructive remarks that have remarkably improved our manuscript. For Q2, we examined not only the NCS loci but also genomic loci that encode other major BIA biosynthetic enzymes like OMT, NMT, and CYP450. For example, in Figure 2D, the focal locus (a biosynthetic gene cluster) comprises OMT and CYP genes. To more fully understand BIA, and specifically bisBIA, biosynthesis in these species, we conducted a series of enzyme experiments to examine the function of these genes. This ongoing biochemical project has the potential to uncover the role of biosynthetic genes and their genomic loci in creating chemical diversity. We hope we could share our findings soon.

Reviewer #2 (Comments to the Author):

The authors did not respond to the reviewer's comments, especially the infection experiment using SARS-CoV-2 live virus instead of VLP (Q2) that is essential for supporting the authors' conclusion. Experimental results obtained with VLP has to be confirmed using SARS-CoV-2 live virus since the VLP is an artificial system as the reviewer commented, but the authors did not answer to the comment. As well, the authors did not improve the description of manuscript in which they still claim "broad-spectrum anti-coronavirus activity" without any experiment using SARS-CoV-1 and MERS-CoV, neglecting the reviewer's comment (Q6). They did not sufficiently improve the inconsistent results between figures (Q4, 5).

The manuscript was not sufficiently improved to be able to be published in the journal.

R2.Q1: Experimental results obtained with VLP has to be confirmed using SARS-CoV-2 live virus.

Response: Thanks for your constructive suggestion. We have followed your suggestions and confirmed the anti-SARS-CoV-2 activity of benzyloquinoline alkaloids using SARS-CoV-2 live virus (SARS-CoV-2/WH-09/human/2020/CHN; GenBank: MT093631.2) in biosafety level 3 lab, and related descriptions in the revised manuscript are as follow:

“The anti-SARS-CoV-2 activities of these eight compounds were further confirmed using live SARS-CoV-2 virus (SARS-CoV-2/WH-09/human/2020/CHN; GenBank: MT093631.2) in biosafety level 3 lab. Seven compounds inhibited SARS-CoV-2 infection in a dose-dependent manner (and dauricine can inhibit SARS-CoV-2 wild type, delta, omicron strains with an EC₅₀ range of 18.2 to 33.3 μM in the Vero E6 cells), with EC₅₀ ranging from 2.19 to 19.95 μM. Among them, cepharanthine was also the most effective compound against SARS-CoV-2, with an EC₅₀ of 2.19 μM, which was consistent with that of remdesivir.” (Line 400-408, page 18)

R2.Q2: the authors did not improve the description of manuscript in which they still claim "broad-spectrum anti-coronavirus activity" without any experiment using SARS-CoV-1 and MERS-CoV, neglecting the reviewer's comment (Q6)

Response: Thank you for your insightful comment. Following your advice, we changed the term “broad-spectrum antiviral activity” to “broad-spectrum anti-coronavirus activity” in the last version

of the manuscript, and "broad-spectrum anti-coronavirus activity" was confirmed in live coronaviruses including SARS-CoV-2 (a member of the genus *Betacoronavirus*), pangolin coronavirus GX_P2V (a member of the genus *Betacoronavirus*), Porcine epidemic diarrhea virus (PEDV, a member of the genus *Alphacoronavirus*), and swine acute diarrhea syndrome coronavirus (SADS-CoV, a member of the genus *Alphacoronavirus*), since experimental condition limitation and law limitation in the mainland of China, "broad-spectrum anti-coronavirus activity" on SARS-CoV-1 and MERS-CoV were performed based on SARS-CoV-1 and MERS-CoV pseudoviruses (Figure 4). Thanks for this insightful comment again.

R2.Q3: They did not sufficiently improve the inconsistent results between figures (Q4, 5)

Response: Thank you for your insightful comment. In the last reply comment, we repeated the inconsistent data you mentioned and found that our data is repeatable. We explained the differences in study methods as a possible reason for data inconsistencies and cited relevant studies. Thanks for this insightful comment again.

Response to reviewer #3:

The revised manuscript has addressed a few minor concerns of mine, including adding discussion on BIA inhibition mechanism and performing NCS assays, as well as the new NCS gene duplication discovery. NCS is the well-known gateway enzyme for BIA. I was hoping that the authors could make new compounds with the putative genes from the genome, to show the use of it. The authors also removed their claim on evo-chemo strategy for accelerating drug discovery.

The two individual components of this manuscript, namely 1) genome assembly and small discovery of NCS duplication, 2) anti-SARS-Cov2 assays with cephalanthine/derivatives etc. are separately sound, and separately useful to researchers in two different fields. Yet the authors tried hard to promote the idea that the genome assembly has guided and been useful for their discovery of other anti-SARS-cov2 compounds. Still there has been no support from this manuscript on this matter, and it remains strange to marry this two components in this manuscript. Clearly the genome mining has not accelerated anti-SARS-Cov2 drug discovery in this article, nor has the authors created new, superior intermediates or derivatives using their new genes, benefiting from the genome assembly. Publication of this work requires substantial new data to show that the genome mining has led to the discovery of new genes, which have led to the biosynthesis of unknown or undetected BIAs superior to cephalanthine. Alternatively, the author may consider separating the two components in two articles.

Response: Thank you for your insightful comments. We agree with your thoughts on discovering new bisBIA compounds through genome mining and biochemical assays.

Our manuscript proposes that mining cepharanthine analogs guided by the *Stephania* genome accelerates the discovery of anti-coronavirus drugs. We assessed the antiviral potential of 23 compounds from the cepharanthine biosynthetic pathway, including dopamine, 12 BIA monomers, and 10 cepharanthine analogs, based on the predicted bisBIA biosynthesis. The results showed that seven bisBIAs had the potential to inhibit coronavirus broadly. Due to their potent antiviral activities, the *Stephania* genome provides important insights into the resource usage and metabolic engineering of bisBIAs. However, given the unknown catalytic enzymes involved in the biosynthesis of bisBIAs, including cepharanthine and its analogs, our study has predicted a few OMT, NMT, and CYP450

genes that may be involved in the methylation, hydroxylation, and C-O phenol coupling. We are currently working on elucidating the catalytic functions and completing the biosynthetic pathway, but this is a complex biochemical project, and we hope to share the results in the future.

Thank you for bringing up your concerns about the new intermediates or derivatives of cepharanthine. And our study did not focus on the modification and biosynthesis of unnatural bisBIAs using the *Stephania* genes. Our study was mainly concerned with the biosynthesis and diversity of bisBIAs in these lineages and the antiviral activities of natural bisBIAs. We completely agree with your suggestion to design and modify natural products to produce superior intermediates or derivatives through green chemistry and metabolic engineering. We have already cloned the candidate OMT, NMTs, and CYP450s, and are presently identifying their biochemical functions. While the biomodification of unnatural products and their antiviral activities are not the main focus of this study, your constructive suggestions do provide important strategies for drug discovery using genome data.

In this study, we aim to explore the significance of genome data in drug discovery. Natural products in various plant lineages have structural diversity that helps them withstand environmental stress during evolution. In addition, bisBIAs are a group of diverse chemicals that can inhibit coronavirus infection. We believe that studying the genetic basis and evolutionary trajectory of active compound biosynthesis, during the plant-vs.-pathogen arms race, can lead to more efficient drug discovery, similar to the human-vs.-pathogen arms race. Additionally, the genome data can provide crucial gene elements for the metabolic engineering of potent bisBIAs. Thanks for this insightful comment again.

Reviewer #4 (Comments to the Author):

I am pleased to see that the authors have addressed all my concerns. I am still not confident about the genome assembly's accuracy based on viewing the HiC contact map. The authors provided enough evidence to say that these are likely not assembly errors when explaining chr1 of *S. japonica*.

Response: Thanks for your positive comments on our manuscript. We have followed your suggestions to improve the manuscript quality.

R2.Q1: validation of each chromosome

But then, when I check their HiC contact map for *S. yunnanensis*, I can see the 3rd box telomere region (from the top) has strange contact with 9th box putative centromere region. If I had this assembly, I would correct this issue, cut this region, and then move it down to 9th box as most likely, this is repeats, but telomeric and centromere regions should be quite distinct. As I mentioned in my previous comments, this aspect requires more careful validation for each chromosome. Once a genome assembly is established, there is little incentive to correct it or validate its correction, and therefore, I feel that it is important to check each chromosome for a genome assembly before it is made public. In my opinion, this remains an issue for me.

Response: We thank you sincerely for your valuable advice, and we agree with your comment that “it is important to check each chromosome for a genome assembly before it is made public”. Following this suggestion, we conducted a new round of manual correction on all three assemblies, carefully checked and validated each chromosome. Table R1 listed dozens of modifications made in this round.

Table R1. Modifications in three species.

Species	Chromosome	Coordinate before correction	Coordinate after correction
	chr1	chr1:51631985-52167410	Scaffold49:1-535426
	chr1	chr1:52167511-53237358	chr1:51631985-52701832
	chr1	chr1:53237459-62284599	chr1:52701933-61749073
Sjap	chr1	chr1:77171942-86876748	chr1:76636416-86341222
	chr1	chr1:62284700-77171841	chr1:61749174-76636315
	chr5	chr5:715159-26304620	chr5:1-25589462

chr5	chr5:1-715058	Scaffold43:1-715058
chr5	chr5:26304721-54028792	chr5:25589563-53313634
chr7	chr7:30671954-43705308	chr7:29120134-42153488
chr7	chr7:44261648-53428141	chr7:42709828-51876321
chr7	chr7:1-620472	Scaffold44:1-620472
chr7	chr7:29229436-30671853	chr7:27677616-29120033
chr7	chr7:43705409-44261547	chr7:42153589-42709727
chr7	chr7:10780459-24581160	Scaffold40:1-446274
chr7	chr7:11401031-25201732	chr7:10780459-24581160
chr7	chr7:25201833-25617023	Scaffold45:1-415191
chr7	chr7:25617124-26132980	Scaffold46:1-515857
chr7	chr7:26133081-29229335	chr7:24581261-27677515
chr7	chr7:620573-11400930	chr7:1-10780358
chr11	chr11:40874560-41263566	Scaffold47:1-389007

chr7	chr7:42383191-42676159	Scaffold6:1-292969
chr7	chr7:42676260-44488029	chr7:42383191-44194960
chr7	chr7:54650138-55793034	chr7:54357069-55499965
chr7	chr7:55793135-56678241	chr7:55500066-56385172
chr9	chr9:11705183-15852202	chr9:11165521-15312540
chr9	chr9:15852303-16614066	chr9:15312641-16074404
chr9	chr9:16614167-44915730	chr9:16074505-44376068
chr9	chr9:44915831-46002068	chr9:44376169-45462406
Scep chr9	chr9:46002169-50683552	chr9:45462507-50143890
chr9	chr9:50683653-51655775	chr9:50143991-51116113
chr9	chr9:51655876-52320462	chr9:51116214-51780800
chr9	chr9:52320563-52738977	chr9:51780901-52199315
chr9	chr9:52739078-53972931	chr9:52199416-53433269
chr9	chr9:7405412-7944973	Scaffold7:1-539562
chr9	chr9:7945074-9909798	chr9:7405412-9370136
chr9	chr9:9909899-10452213	chr9:9370237-9912551
chr10	chr10:31497701-31670017	Scaffold8:1-172317

chr10	chr10:31670118-38111789	chr10:31497701-37939372
chr10	chr10:38111890-39200552	chr10:37939473-39028135
chr10	chr10:39200653-42321439	chr10:39028236-42149022
chr10	chr10:42321540-44323144	chr10:42149123-44150727
chr10	chr10:44323245-45728076	chr10:44150828-45555659
chr10	chr10:45728177-46921576	chr10:45555760-46749159
chr10	chr10:48057490-48238438	chr10:47885073-48066021
chr10	chr10:49819688-52358023	chr10:49647271-52185606
chr13	chr13:15098857-15621183	chr13:14587797-15110123
chr13	chr13:15621284-19051816	chr13:15110224-18540756
chr13	chr13:43031927-45188691	chr13:42520867-44677631
chr13	chr13:45188792-48007401	chr13:44677732-47496341
chr13	chr13:7152482-7663441	Scaffold9:1-510960
chr13	chr13:7663542-8589572	chr13:7152482-8078512
		
chr2	chr2:32316820-32620548	Scaffold51:1-303729
chr2	chr2:32620649-68962812	chr2:32316820-68658983
chr3	chr3:42242458-43339060	Scaffold52:1-1096603
chr3	chr3:43339161-54029112	chr3:42242458-52932409
chr3	chr3:54029213-65793035	chr3:52932510-64696332
chr5	chr5:24702849-25257962	Scaffold50:1-555114
Syun	chr5:25258063-25666690	chr5:24702849-25111476
	chr5:25666791-63272587	chr5:25111577-62717373
chr8	chr8:41648416-42047454	Scaffold49:1-399039
chr8	chr8:42047555-45047941	chr8:41648416-44648802
chr8	chr8:45048042-46497350	chr8:44648903-46098211
chr8	chr8:46497451-47992131	chr8:46098312-47592992
chr8	chr8:47992232-57109000	chr8:47593093-56709861

Sjap: *Stephania japonica*; Scep: *Stephania cepharantha*; Syun: *Stephania yunnanensis*.

In this revision, we mainly checked short contigs made mainly of repetitive sequences. These contigs are usually placed in peri- centromere or telomere regions and may potentially incorrectly link to

other regions that are enriched in similar repeats. It is worth mentioning that we followed the same criteria to conduct the manual correction in all three species. We visualized the HiC contact maps of three species and compared them with the ones before correction (Figure R1). We observe satisfactory improvement after this round for *Stephania yunnanensis* and *Stephania cepharantha*. For instance, the incorrect contact between Chr3 and Chr9 in *Stephania yunnanensis* has been corrected. However, the correction seems to bring in some problems in *Stephania japonica* (Figure R1-A/D). This discrepancy reflects the complexity of genome assembly manual correction. Even for closely related species, a correction strategy may perform well in some species but fail in others. Upon thorough evaluation of the findings, it was determined that the previous assembly for *Stephania japonica*, the current assemblies for *Stephania yunnanensis* and *Stephania cepharantha* should be designated as the final assemblies. Figure 1B has been revised accordingly. (Line 240-241, page 11)

Figure R1. HiC contact map before and after a new round of manual correction. A-C, contact maps of *Stephania japonica*, *Stephania yunnanensis*, and *Stephania cepharantha* before correction; D-F, contact maps of *Stephania japonica*, *Stephania yunnanensis*, and *Stephania cepharantha* after correction.

R2.Q2:

Other than that, authors have done a great job of improving the text and added more information. Lastly, I request authors to please include all your scripts that were used for creating figures, analysis, TAD analysis, gene cluster analysis and so on. This is important for open science, and I am not able to find scripts used for genome assembly and comparative genome analysis in the mentioned github.

Response: We apologize for this inadvertent oversight. We added all my scripts that were used for creating figures, analysis, TAD analysis, and gene cluster analysis on the github repository (https://github.com/liuzy2008/evo-chemo_anti-SARS-CoV-2_drug_discovery2023/tree/main/scripts).

Figure R2 is a snapshot of the corresponding page in github.

Figure R2. A snapshot of github page storing scripts.

Reviewers' Comments:

Reviewer #1:

Remarks to the Author:

I acknowledge deeply for the efforts that the authors have made by taking all the reviewers' comments to improve this study. I revisited and went over the whole revised manuscript; however, I still found this study was lack of novelty except high-quality genome assemblies and discovery of cepharanthine analogs with anti-coronavirus activities.

The major concern still goes to the proposed search strategy (figure 7) based on genome-mining. However, the authors actually have used untargeted metabolomics rather than genome-mining to have identified the metabolites for drug discovery (L316-317). Putative cepharanthine biosynthesis pathway genes were proposed based on well-studied BIA biosynthesis orthologous genes. Transcriptome data can be applied to do the same job, weakening the importance of genome assembly in this study. In addition, neither new genes nor gene clusters (which do require genome information) were functionally identified. I suggest the authors to separate the genome and the drug-activity parts into two stories for publications.

Reviewer #2:

Remarks to the Author:

The authors appropriately responded to the reviewer's comments with adding the experiments using live SARS-CoV-2 virus and the manuscript is now worth publishing.

Reviewer #3:

Remarks to the Author:

I appreciate the authors' enthusiasm on their work involving a new *Stephania* genome assembly and meticulous testing of a number of bis-BIAs for their antiviral activities. Both studies hold significance for the scientific community, and the manuscript has undergone notable improvements by addressing unfounded claims. With the respect of lacking coherence between the genomic and antiviral aspects of the work, I defer this matter to the editor for a decision.

I recommend that the authors consider altering the title to avoid potential misinterpretation. The current title may lead readers to believe that the cepharanthine analog mining in this study is guided by the genome, which is not the case. A revised title such as "Cepharanthine Analogs Mining and a Chromosomal Level-Genome in *Stephania* Accelerate Anti-Coronavirus Drug Discovery" or a similar version might more accurately reflect the content of the study.

Reviewer #4:

Remarks to the Author:

In the revised manuscript, titled, "Genome-guided Cepharanthine analogs mining in *Stephania* accelerates 1 anti-coronavirus drug discovery", authors have now addressed all of my concerns. I am happy to agree that the genome assembly quality is good, which was one of my major concern. Therefore, I support this article for publication at Nature communication.

REVIEWERS' COMMENTS

Reviewer #1 (Remarks to the Author):

I acknowledge deeply for the efforts that the authors have made by taking all the reviewers' comments to improve this study. I revisited and went over the whole revised manuscript; however, I still found this study was lack of novelty except high-quality genome assemblies and discovery of cepharanthine analogs with anti-coronavirus activities.

The major concern still goes to the proposed search strategy (figure 7) based on genome-mining. However, the authors actually have used untargeted metabolomics rather than genome-mining to have identified the metabolites for drug discovery (L316-317). Putative cepharanthine biosynthesis pathway genes were proposed based on well-studied BIA biosynthesis orthologous genes. Transcriptome data can be applied to do the same job, weakening the importance of genome assembly in this study. In addition, neither new genes nor gene clusters (which do require genome information) were functionally identified. I suggest the authors to separate the genome and the drug-activity parts into two stories for publications.

Response: Thank you very much for your constructive comments. Following your and the Editors' advice, we have removed the proposed strategy (Figure 7) to avoid declaring that the cepharanthine analog mining in this study is guided by the genome.

Reviewer #2 (Remarks to the Author):

The authors appropriately responded to the reviewer's comments with adding the experiments using live SARS-CoV-2 virus and the manuscript is now worth publishing.

Response: Thank you for your hard work in enhancing our manuscript. We appreciate your efforts.

Reviewer #3 (Remarks to the Author):

I appreciate the authors' enthusiasm on their work involving a new *Stephania* genome assembly and meticulous testing of a number of bis-BIAs for their antiviral activities. Both studies hold significance for the scientific community, and the manuscript has undergone notable improvements by addressing unfounded claims. With the respect of lacking coherence between the genomic and antiviral aspects of the work, I defer this matter to the editor for a decision.

I recommend that the authors consider altering the title to avoid potential misinterpretation. The current title may lead readers to believe that the cepharanthine analog mining in this study is guided by the genome, which is not the case. A revised title such as "Cepharanthine Analogs Mining and a Chromosomal Level-Genome in *Stephania* Accelerate Anti-Coronavirus Drug Discovery" or a similar version might more accurately reflect the content of the study.

Response: Thank you very much for your positive comments. We appreciate your efforts in improving the quality of our manuscript. Following your and the Editors' advice, we have removed the proposed strategy (Figure 7) to avoid declaring that the cepharanthine analog mining in this study is guided by the genome. In addition, we have also revised the title to "Cepharanthine analogs mining and genomes of *Stephania* accelerate anti-coronavirus drug discovery".

Reviewer #4 (Remarks to the Author):

In the revised manuscript, titled, "Genome-guided Cepharanthine analogs mining in *Stephania* accelerates 1 anti-coronavirus drug discovery", authors have now addressed all of my concerns. I am happy to agree that the genome assembly quality is good, which was one of my major concern. Therefore, I support this article for publication at Nature communication.

Response: Thank you very much for your efforts in improving our manuscript.